# Ultrafast visualization of incipient plasticity in dynamically compressed matter

Mianzhen Mo [1,8✉], Minxue Tang [2,8], Zhijiang Chen [1], J. Ryan Peterson[1,3], Xiaozhe Shen [1], John Kevin Baldwin[4], Mungo Frost [1], Mike Kozina[1], Alexander Reid [1], Yongqiang Wang [4,5], Juncheng E [6], Adrien Descamps [1,7], Benjamin K. Ofori-Okai [1], Renkai Li [1], Sheng-Nian Luo [2✉], Xijie Wang [1✉] & Siegfried Glenzer [1✉]

Plasticity is ubiquitous and plays a critical role in material deformation and damage; it inherently involves the atomistic length scale and picosecond time scale. A fundamental understanding of the elastic-plastic deformation transition, in particular, incipient plasticity, has been a grand challenge in high-pressure and high-strain-rate environments, impeded largely by experimental limitations on spatial and temporal resolution. Here, we report femtosecond MeV electron diffraction measurements visualizing the three-dimensional (3D) response of single-crystal aluminum to the ultrafast laser-induced compression. We capture lattice transitioning from a purely elastic to a plastically relaxed state within 5 ps, after reaching an elastic limit of ~25 GPa. Our results allow the direct determination of dislocation nucleation and transport that constitute the underlying defect kinetics of incipient plasticity. Large-scale molecular dynamics simulations show good agreement with the experiment and provide an atomic-level description of the dislocation-mediated plasticity.

[1] SLAC National Accelerator Laboratory, Menlo Park, CA 94025, USA. [2] School of Materials Science and Engineering, Southwest Jiaotong University, Chengdu, Sichuan 610031, P. R. China. [3] Physics Department, Stanford University, Stanford, CA 94305, USA. [4] Center for Integrated Nanotechnologies, Los Alamos National Laboratory, Los Alamos, NM 87545, USA. [5] Materials Science and Technology Division, Los Alamos National Laboratory, Los Alamos, NM 87545, USA. [6] European XFEL GmbH, 22869 Schenefeld, Germany. [7] Aeronautics and Astronautics Department, Stanford University, Stanford, CA 94305, USA. [8]These authors contributed equally: Mianzhen Mo, Minxue Tang. ✉email: mmo09@slac.stanford.edu; sluo@swjtu.edu.cn; wangxj@slac.stanford.edu; glenzer@slac.stanford.edu

Dynamic compression waves propagating into solids can cause irreversible plastic deformation where the deviatoric stresses exceed yield strengths. A fundamental understanding of this deformation is crucial for a broad range of applications, including planetary formation and meteor impacts[1,2], laser material processing[3], high-performance ceramics[4], and inertial confinement fusion experiments[5]. At the lattice level, plastic deformation in crystalline materials is intimately related to dislocations (line defects) whose motions result in crystal slippage along lattice planes. The subsequent interaction and multiplication of these dislocations give rise to plastic flow inside the material and renders it evolve towards a three-dimensional (3D) hydrostatic state, indicating the importance of dislocation dynamics in plastic deformation.

Despite intense investigations, experimental visualization and detailed understanding of dislocation dynamics during plastic deformation at high strain rates are still lacking. One key challenge is to determine the dislocation nucleation and transport processes at the early stage of plasticity[6,7], i.e. the incipient plasticity. Observing these processes requires a probe with a lattice-level spatial resolution (Å) and picosecond temporal resolution. Conventional velocimetry techniques that measure the elastic and plastic waves, however, cannot resolve the underlying lattice dynamics[8,9]. On the other hand, in situ time-resolved diffraction techniques directly measure the microscopic lattice response and therefore have been widely used to study plastic deformation. Early diffraction experiments using nanosecond-resolution X-rays showed a prompt 1D to 3D lattice relaxation but lacked the temporal resolution to resolve the dislocation dynamics[10–13]. More recently, X-ray free-electron laser (XFEL)-based diffraction experiments on laser-shocked polycrystalline materials had overcome the limitation on temporal resolution and captured the 3D lattice relaxation within a few tens of picoseconds[14]. However, the inherent powder diffraction nature of polycrystalline materials averages out the directionality of lattice deformation, which is critical to reveal lattice-level deformation mechanisms[15]; single-crystal diffraction is more suitable in this regard. High spatio-temporal resolution measurements of lattice dynamics in single crystals are urgently needed to capture the incipient plasticity and to validate theoretical predictions of dislocation dynamics during high strain-rate dynamic loading.

It is within this context that we performed an ultrafast, in situ time-resolved diffraction experiment using MeV electrons to resolve the lattice dynamics of single-crystal aluminum (Al) under laser-driven compression at strain rates of ~$10^9$ per second. The flat Ewald sphere of MeV electrons allows for a large-volume sampling of the reciprocal space and the simultaneous capture of multiple orders of diffraction spots, enabling a "3D view" of lattice deformation as the compression wave propagates through the sample. In the meantime, a 600-fs-long bunch length of the relativistic electrons (~220 μm in diameter) makes it possible to probe the ultrafast microstructural evolution during plastic deformation. Using this time-resolved electron diffraction technique, our experiment demonstrates a complete history of incipient plasticity in single-crystal materials. These results reveal the lattice transitioning from a 1D elastic to a 3D plastic state within ~5 ps after reaching the elastic limit of ~25 GPa. Furthermore, the experimental data allows us to accurately determine the dislocation nucleation and transport processes. Our experiments are compared directly with molecular dynamics (MD) simulations performed at similar length and temporal scales. MD simulations show good agreement with experiments and corroborate our observation of the dislocation-mediated plasticity. Our experimental results combined with MD simulations provide atomic-level insights into the dislocation dynamics, revealing their nucleation and transport properties.

## Results and discussion

**Experimental setup and static diffraction signal.** This experiment was performed on the mega-electronvolt ultrafast electron diffraction (MeV-UED) instrument of the Linac Coherent Light Source (LCLS)[16,17]. The time-resolved single-shot electron diffraction measurements were performed in the transmission geometry with 600-fs-long electron pulses at 3.7 MeV measuring the lattice responses up to 70 ps of delay time (Fig. 1a). The samples used in our experiments were 200-nm-thick, free-standing, (110)-oriented, single-crystal Al thin films mounted on 500 μm × 500 μm apertures. The single-crystal samples were rotated by a fix angle about the vertical axis to probe the desired lattice planes (Fig. 1b). Compressive ramp waves were generated along the normal direction of the rear sample surface via ablation pressure with a 800 nm, 20 ps [full width at half maximum (FWHM)], 10 mJ laser pulse. The drive laser spot on the sample had a Gaussian intensity profile with a FWHM of ~360 μm. The considerably large ratio of the pump laser spot size over the sample thickness ensured that the edge release effect can be neglected for the time duration of the experiments[18]. Furthermore, the application of ramp-wave loading in our experiments provided a relatively prolonged elastic–plastic transition, facilitating the time-resolved UED measurement.

Our pump-probe geometry captured both the lattice deformation parallel and perpendicular to the loading direction. Figure 1b illustrates this from the unit lattice perspective, along with simulated and measured diffraction patterns of the uncompressed single-crystal Al. For 1D purely elastic deformation, lattice planes will not deform when parallel to the loading or longitudinal direction, i.e., only those planes intersecting the loading axis deform, leading to a corresponding shift of the Laue diffraction peaks along the $\pm \mathbf{k}_x$ directions in the detector plane. For 3D plastic lattice deformation, lattice planes parallel to the loading direction will also be affected and the corresponding Laue peaks will shift along the $\pm \mathbf{k}_y$ directions. In our experiments, we used the {220}-family peaks to track the transverse and longitudinal elastic strains. The distance between Laue peaks ($2\bar{2}0$) and ($\bar{2}20$), can be used to quantify the transverse elastic strain, whereas the distances of the other two pairs of centrosymmetric peaks are for the longitudinal elastic strain (Materials and methods are available as supplementary materials). For convenience, we refer to the former peak pairs as the transverse strain peak pair (TSPP) and the latter as the longitudinal strain peak pairs (LSPPs). See Supplementary Fig. 3a for more details (Materials and methods are available as supplementary materials). By monitoring the change of the distances between these diffraction peak pairs at different time delays, we can reveal the deformation history including the elastic–plastic transition (Materials and methods are available as supplementary materials).

**Temporal evolution of diffraction data.** Figure 2 shows the diffraction data as the compression wave gains strength and propagates through the sample. Figure 2a presents the raw diffraction patterns of the compressed samples at selective time delays ($t$) for an incident fluence of $4.3 \pm 0.4$ J cm$^{-2}$. Figure 2b and c show the corresponding intensity profiles of the LSPP and the TSPP, respectively, together with Gaussian fits. In the LSPP case (Fig. 2b), the spacing between the two peaks increases with $t$, an indication of increasing lattice compression as the stress wave propagates along the longitudinal direction. However, in the TSPP case, the peak spacing remains unchanged at $t = 22$ ps, but increases with $t$ afterward due to the occurrence of plastic deformation (see below). The peak intensities of the LSPPs show a slight enhancement at early time, followed by an exponential decay afterward (see Supplementary Fig. 7c). This trend is in contrast to the TSPP case where intensity decay begins shortly

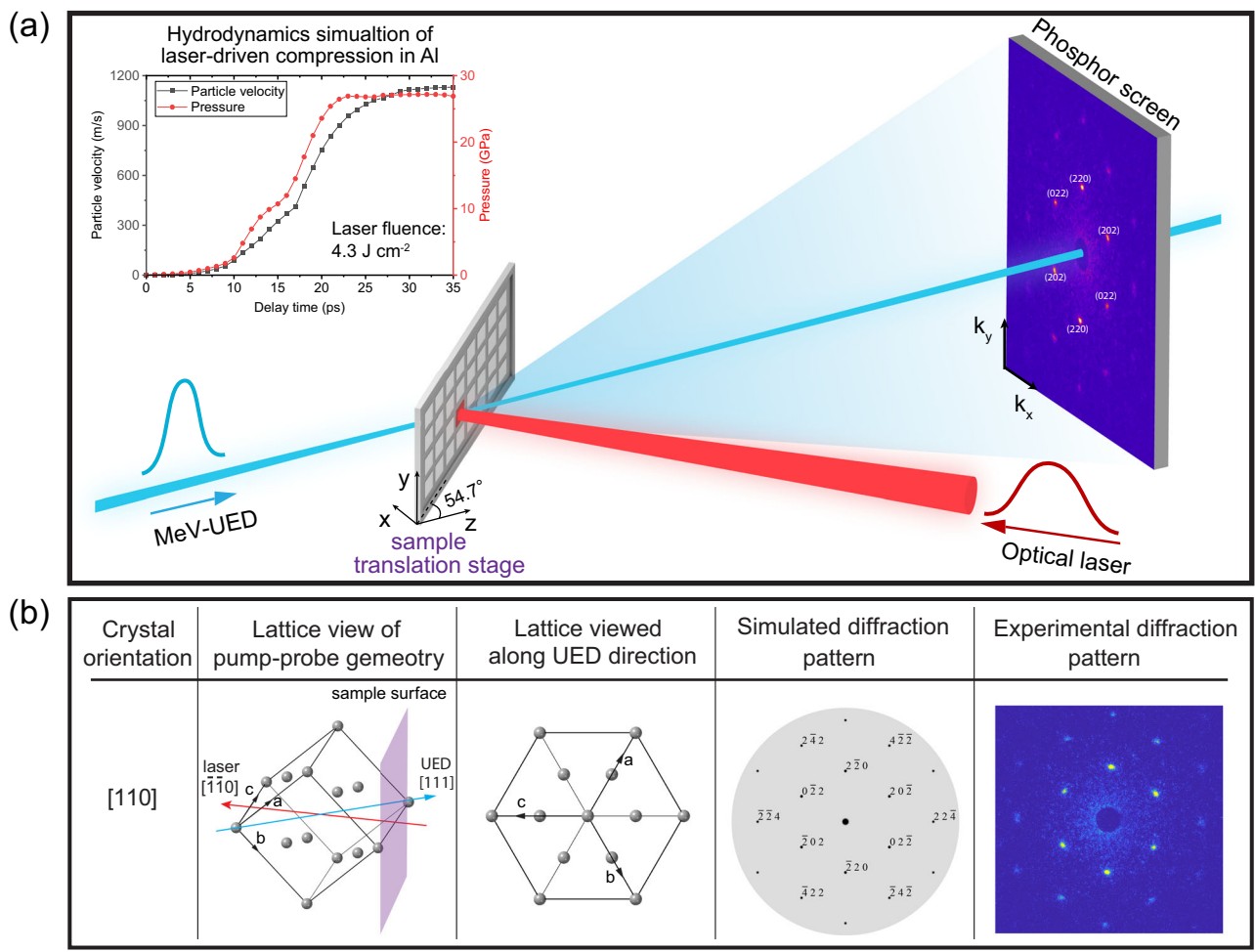

**Fig. 1 Configuration for laser pump−electron probe experiments and diffraction patterns of Al single crystals at ambient (uncompressed) conditions. a** Experimental setup for in situ UED under dynamic compression. The 3.7-MeV electrons generated by a radio-frequency (RF) gun are focused by two separate solenoids onto the sample; the electron beam spot size at the sample location is ~184 μm × 250 μm (*x* × *y*), the electron bunch charge is ~90 fC and the pulse duration is ~0.6 ps. The sample is rotated around the *y*-axis by a certain angle to probe lattice planes with a desired orientation (see bottom panel). The sample is ablated by counter-propagating laser pulses (800 nm, 20 ps FWHM, ≤10 mJ) with a Gaussian-like intensity profile (~360 μm FWHM). The laser incident angle is close to normal incidence. The inset shows the time history of peak particle velocity and peak pressure in 200-nm-thick Al irradiated with a laser fluence of 4.3 J cm$^{-2}$, which are obtained by hydrodynamics simulation (see the "Methods" section). **b** The pump-probe spatial overlap viewed from the crystal lattice perspective, and the comparison of the ambient diffraction patterns between simulations and experiments. The simulated diffraction pattern is used to identify the detected lattice planes.

after the loading starts. This observed increase in diffraction intensity at the early stage for LSPPs is likely due to the re-alignment of some mis-oriented grains caused by the initial compression. The peak widths for both LSPP and TSPP (see Supplementary Fig. 7f), remain unchanged until *t* = 27 ps, and then undergo a progressive increase with *t*. The onset of peak broadening coincides with the onset of plastic deformation (Fig. 3c), implying that the peak width broadening is likely caused by Huang diffuse scattering[19] of defects generated during the plastic deformation.

**Time history of longitudinal and transverse elastic strain**. To further understand the elastic–plastic deformation process, we calculate the longitudinal elastic strain, $\varepsilon_l^e$, and the transverse elastic strain, $\varepsilon_t^e$, from the corresponding diffraction peak shifts (Materials and methods are available as supplementary materials). Figure 3a–c show the time histories of $\varepsilon_l^e$ and $\varepsilon_t^e$ obtained at three different pump fluences, in comparison with the results from MD simulations (see the "Methods" section). At the lowest fluence of 1.2 J cm$^{-2}$ (Fig. 3a), with increasing time delay,

$\varepsilon_l^e$ first increases and then decreases whereas $\varepsilon_t^e$ remains essentially at zero, suggesting a purely elastic 1D deformation over the loading period. The longitudinal elastic strain reaches $\varepsilon_l^e \approx 0.04$ at *t* = 32 ps, and then rapidly decreases. The time for reaching the peak longitudinal strain coincides with the expected transit time for the longitudinal sound wave traversing the whole sample at 6.3 km s$^{-1}$ (ref. [20]). A similar strain evolution is observed for the pump fluence of 2.2 J cm$^{-2}$ (Fig. 3b), except a higher peak strain of $\varepsilon_l^e = 0.067$ is found as a result of a stronger drive.

However, at the highest fluence of 4.3 J cm$^{-2}$ (Fig. 3c), the transverse elastic strain (recast in Fig. 3d) increases during the time interval of 27 ps < *t* < 42 ps to $\varepsilon_t^e = 0.007$ as a result of the dislocation motions. The elastic limit, corresponding to the longitudinal elastic strain at the onset of $\varepsilon_t^e$, is measured to $\varepsilon_l^e = 0.075$. Our measurement of the elastic limit is slightly above the peak longitudinal elastic strain that was reached for a fluence of 2.2 J cm$^{-2}$ (Fig. 3b); consequently, no plasticity is observed in this case. Our observation of the time delay between the rises of $\varepsilon_l^e$ and $\varepsilon_t^e$ and their co-existence demonstrate the capture of the very incipient plasticity and the elastic–plastic two-wave structure[7].

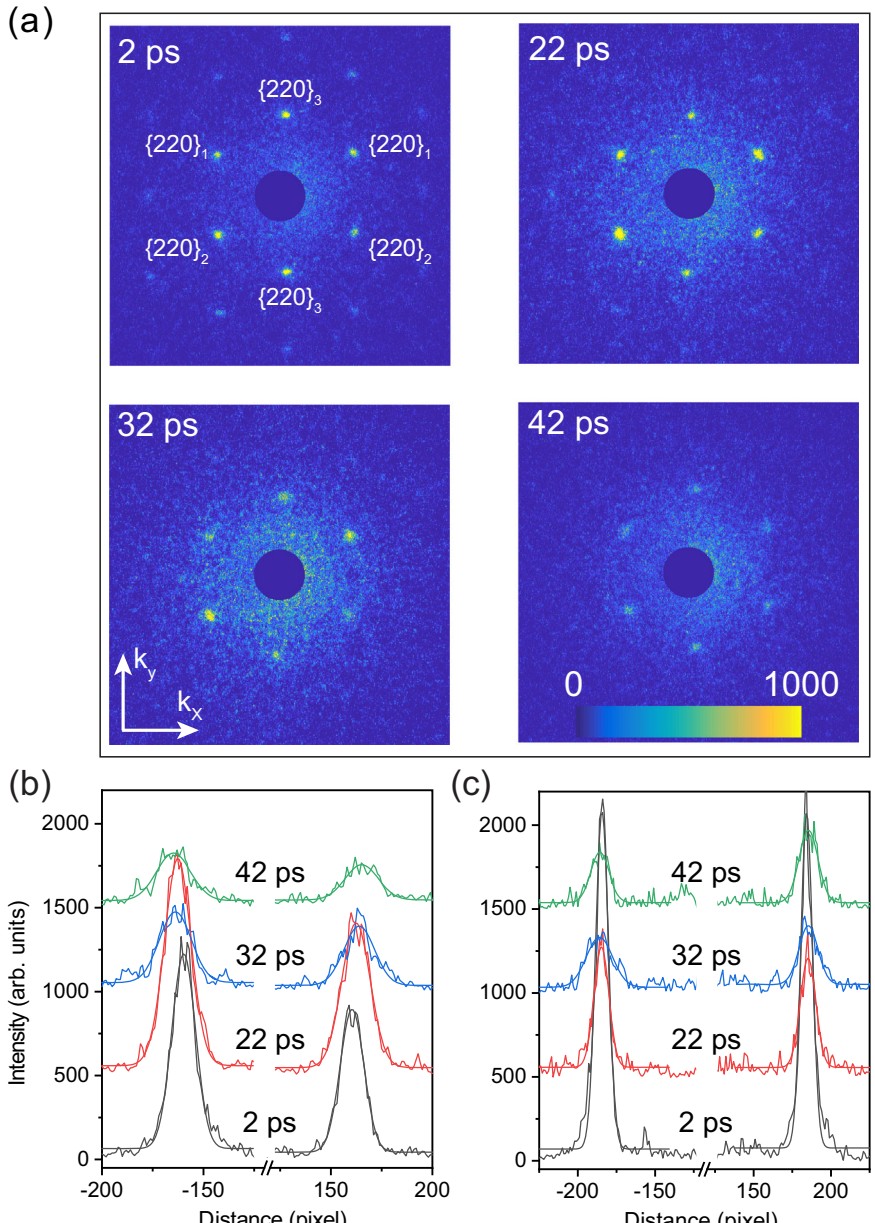

**Fig. 2 Diffraction pattern evolution of (110) single-crystal Al under laser-driven compression at 4.3 ± 0.4 J cm$^{-2}$. a** Single-shot diffraction patterns captured at different pump-probe delays. The same color axes are applied for the images and the color bar represents the scattering intensity in arbitrary units. **b** Intensity lineouts (along **k**$_x$ direction) of the longitudinal strain peak pairs (LSPPs), indicated by {220}$_1$ and {220}$_2$, for the displayed diffraction patterns in (**a**). Here, the average of the two intensity lineouts for {220}$_1$ and {220}$_2$ peak pairs is shown and offset along the ordinate axis for clarity. The origin of the abscissa axis is defined as the center of the two peaks. **c** The same as (**b**) but for the intensity lineouts (along **k**$_y$ direction) of the transverse strain peak pair (TSPP), {220}$_3$. See text and supplementary materials for details.

Note that, upon the plastic deformation, there is a 1D (longitudinal only) to 3D (longitudinal and transverse) transition in lattice deformation. Our experiment suggests that this transition occurs within 5 ps after reaching the elastic limit. The transverse elastic strain continues to grow even after the longitudinal elastic strain reaches its peak. At later time, plastic deformation is interrupted by the rarefaction wave originating from the sample free surface.

These experimental results of the temporal evolution of $\varepsilon_l^e$ and $\varepsilon_t^e$ and their dependence on pump fluence are well reproduced by our MD simulations performed with the experimental conditions. For the simulation of the highest fluence, in addition to $\varepsilon_l^e$ and $\varepsilon_t^e$, we have also calculated the densities of different types of

dislocation defects developed in the compressed sample, including Shockley partial dislocation and perfect dislocation (Fig. 3d). Their temporal behaviors show remarkable agreement with the measured behavior of $\varepsilon_t^e$, therefore corroborating our observation of the dislocation-mediated plasticity. Furthermore, MD simulation suggests that Shockley partial dislocation is the predominant line defect that determines the plastic flow under our experimental conditions. MD simulation also reveals the presence of stacking faults with the generation of Shockley partial dislocations. For example, at $t = 42$ ps, ~8.9% of stacking faults is found in the full simulated structure for the highest fluence case. The effect of these stacking faults on diffraction peak shifts is evaluated and the result indicates that stacking faults will not

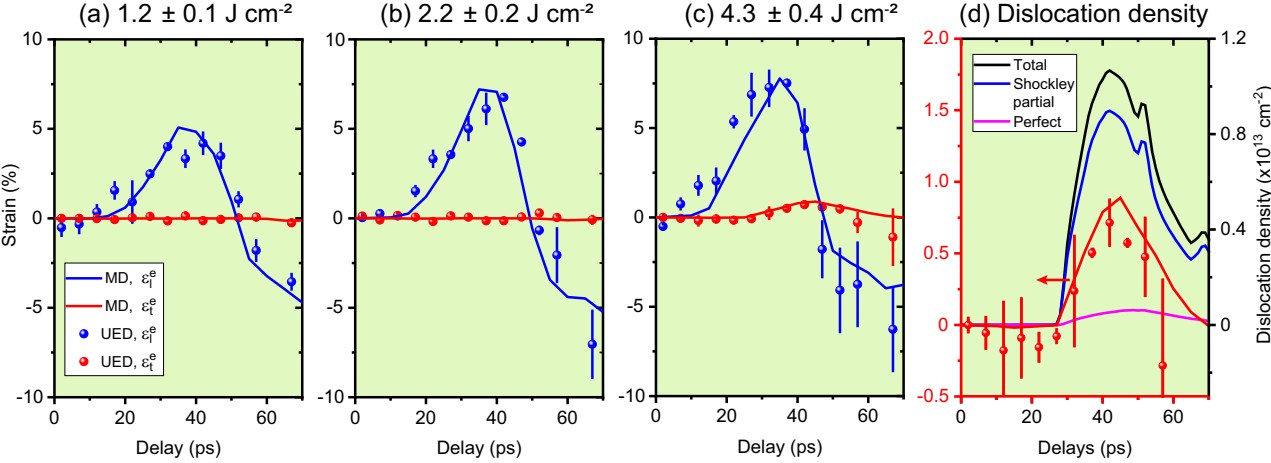

**Fig. 3 Strain evolution of (110) single-crystal Al under laser-driven compression obtained from UED experiments (filled circles) and MD simulations (lines). a–c** are the results at incident laser fluences of 1.2, 2.2, and 4.3 J cm$^{-2}$, respectively. $\varepsilon_i^e$ denotes the longitudinal elastic strain; $\varepsilon_t^e$ denotes the transverse elastic strain. The error bars represent 1 standard deviation uncertainties. The measured evolution of $\varepsilon_t^e$ shown in (**c**) is recast in (**d**), along with the evolution of total, Shockley partial and perfect dislocation densities obtained from MD simulation performed at the same experimental conditions.

affect the {220}-family peaks employed to infer elastic strains in this study (Materials and methods are available as supplementary materials).

**Dislocation dynamics**. From the measured transverse elastic strain (Fig. 3c), we obtained the corresponding plastic strain following $\varepsilon_t^p = -\varepsilon_t^e$ (Materials and methods are available as supplementary materials). The mobile dislocation density, $\rho_m$, can then be deduced through Orowan's equation[21], $\dot{\varepsilon}_t^p = \rho_m v_d b$. Here $\dot{\varepsilon}_t^p$, $v_d$ and $b$ are the plastic strain rate, the average dislocation velocity and the Burgers vector, respectively. Figure 4a shows the results for $\dot{\varepsilon}_t^p(t)$ and $\rho_m(t)$ obtained for $t \le 42$ ps, i.e., before the transverse elastic strain reaches its peak, with $b = 2.86$ Å for (110) single-crystal Al and $v_d = 2.5$ km s$^{-1}$ (ref. [20]). Despite that the average dislocation velocity is not constant over the deformation process, as shown by previous MD simulations[6], the variations in $v_d$ will not change the behavior of $\rho_m(t)$. Thus, these data resolve the phases of the ultrafast material response to high pressure-induced deformation.

The plastic strain rate and the mobile dislocation density evolution (Fig. 4a) feature three distinct stages: (I) 0 ps < $t$ < 22 ps, during this time period, $\rho_m$ is essentially zero, indicating purely elastic deformation. (II) 22 ps < $t$ < 32 ps, here $\rho_m$ monotonically increases, indicating the nucleation and multiplication of dislocation line defects. $\rho_m$ is ~5.5 × 10$^{10}$ cm$^{-2}$ at the onset of plastic deformation (27 ps), and reaches its peak value of ~8.5 × 10$^{10}$ cm$^{-2}$ at 32 ps when the plastic strain rate peaks. (III) 32 ps < $t$ < 42 ps, we observed that $\rho_m$ decreases and approaches zero when the maximum plastic strain is reached. This stage represents the dislocation transport. The peak value of our derived $\rho_m$ is consistent with the previous model estimate of ~10$^{11}$ cm$^{-2}$ (ref. [20]), but it is lower than the total number of dislocation density (10$^{13}$ cm$^{-2}$) obtained from MD simulations, as shown in Fig. 3d. This discrepancy was also seen by previous work on copper[14] and is attributed to the methods commonly employed in MD simulations to extract the dislocation densities, which result in a total number of dislocations to be artificially higher than the number of mobile dislocations responsible for plastic flow.

To further understand the dislocation dynamics, we examine the atomic trajectories from MD simulations using the LAMMPS code[22] (see the "Methods" section). Figure 4b presents a series of snapshots of the dislocation evolution in a selected region of the simulation box. These results show that the onset of plasticity at

the elastic limit (~25 GPa) occurs at $t \approx 27$ ps, in excellent agreement with the experimental observation. At 27 ps, while a finite mobile dislocation density is found, the nearly absence of dislocation lines implies the nucleation of a significant amount of dislocation embryos (Stage I). As stress builds up, the increasing shear stress leads to the activation of more active slip systems and subsequently the increase of dislocation lines (Stage II). Meanwhile, dislocation obstructions can accumulate and slow down this process (Stage III). Under our experimental condition, the time window for the unobstructed dislocation nucleation is relatively short, ~5 ps after the onset of plasticity, in part due to the limited sample dimensions. Dislocation density peaks at ~42 ps in MD simulations as a result of dislocation entanglement and subsequent rarefaction wave from the sample free surface, and the number of mobile dislocations is reduced considerably as in Fig. 4a.

Similar evolution processes of dislocations occur asynchronously across the compressed sample, leading to the inhomogeneity of the overall dislocation structure. An example is shown by Fig. 4c that presents the snapshot of the overall dislocation structure captured at $t = 35$ ps. The corresponding shear stress profile is illustrated in Fig. 4d with a clear separation between the elastic and plastic zones. The elastic shock zone starts behind the front of the ramp wave and extends to a length of ~80 nm until the plastic zone is reached. In the elastic shock zone, the shear stress profile is featured by a quasi-plateau at ~14 GPa and no dislocation is found. As it transitions to the plastic zone, the nucleation of dislocations immediately releases the shear stress; the propagation and multiplication of dislocations lead to a further decline of the shear stress; when dislocations are entangled and saturated, the shear stress or residual yield strength is leveled at ~1.8 GPa, implying that the lattice compression is approaching a fully 3D hydrostatic state.

In summary, our ultrafast, time-resolved electron diffraction measurements provide a lattice-level view of the incipient plasticity dynamics in a widely used metal under ultrahigh strain-rate conditions, via direct measurements of the plastic strain. MeV-UED combined with laser-driven compression provides unprecedentedly the ultrahigh temporal and spatial resolutions, and the broad reciprocal space sampling to resolve the structural dynamics at the very onset of plastic deformation. These results resolve unambiguously the associated 1D to 3D lattice deformation processes and provide a complete history of

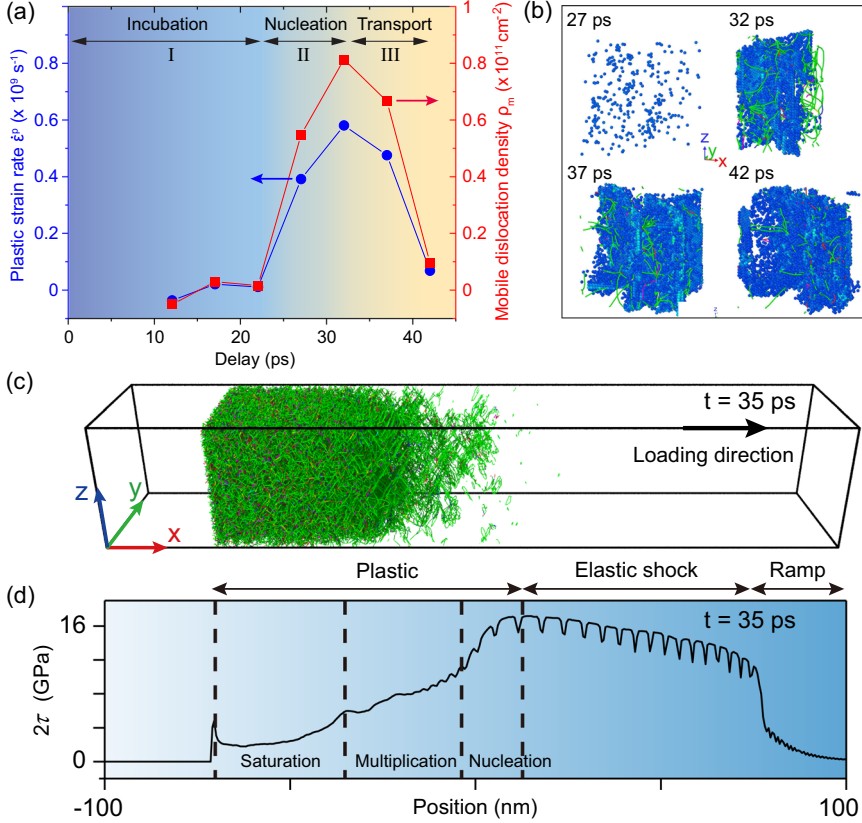

**Fig. 4 Dislocation dynamics of (110) single-crystal Al under laser-driven compression at 4.3 ± 0.4 J cm$^{-2}$. a** Plastic strain rate ($\dot{\varepsilon}^p$; blue line + dots) and mobile dislocation density ($\rho_m$; red line + squares) from UED experiments. Stages I–III correspond to incubation (no dislocations), dislocation nucleation, and dislocation transport, respectively. **b–d** are MD simulation results obtained at the fluence of 4.3 J cm$^{-2}$; **b** Snapshots of atomic configuration at a selected cubical region with 8-nm-long edges, showing dislocation nucleation and transport during plastic deformation. Lines of different colors represent different types of dislocations, with Shockley partial dislocations in green and perfect dislocations in blue. Blue dots are atoms with finite von Mises shear strains. **c** Overall dislocation structure obtained at a time delay of 35 ps; the size of the y–z cross section is 35 nm × 35 nm; the same color coding as in (**b**) is applied for the displayed dislocation lines. **d** the shear stress ($2\tau$) profile of (**c**), showing three regions of the stress relaxation with the dislocation evolution dominated by nucleation, multiplication (transport), and saturation.

incipient plasticity in single-crystal materials. Our results also address the long-standing problem of resolving the dislocation dynamics that is critical to predict dynamic material strength in extreme conditions. Our experiments are compared directly with molecular dynamics simulations performed at similar length and temporal scales. The good agreement between experiments and simulations provides a full atomic-level picture of the ultrafast elastic–plastic deformation in single-crystal materials. The techniques demonstrated here will open up a new horizon for investigating a broad range of high-pressure and high strain-rate phenomena[23–25], including dynamic plastic deformation[26] and phase transitions[27,28].

## Methods

**Experimental details**. The pump-probe experiments were performed in the Accelerator Structure Test Area (ASTA) facility at SLAC National Accelerator Laboratory[16,17]. The MeV electrons for diffraction were achieved by a LCLS-type photocathode radio-frequency (RF) gun. The RF gun is powered by a pulse-forming-network based modulator and a 50-MW S-band klystron. Time-resolved diffraction experiments were performed in the transmission geometry with relativistic electrons at kinetic energies of 3.7 MeV. The relativistic electrons are focused by two separated solenoids onto the target with edge-to-edge diameters of ~184 μm × 250 μm (x × y), bunch charges of ~90 fC and pulse durations of ~0.6 ps at FWHM. The electron detector is located ~3.2 m away from the sample and consists of a P43 phosphor screen, a lens system and a sensitive electron-multiplying CCD (EMCCD) camera (Andor iXon Ultra 888). In the middle of the phosphor screen there is a 1.6-mm-diameter through-hole to prevent the zero-order diffraction signal from saturating the CCD image at high gain during the experiments.

The targets used in our experiments are 200-nm-thick free-standing (110)-oriented single-crystalline Al thin films mounted on 500 μm × 500 μm windows. The sample thickness is slightly less than the elastic mean-free-path (~230 nm) of 3.7 MeV electrons. The misorientation of the Al single-crystal targets was measured to be approximately ±2.5° by rocking the crystal and observing the change of the diffraction pattern. This measured misorientation is very close to the nominal misorientation (±3°) of the NaCl substrates used for fabricating the single-crystal samples.

For the dynamic compression experiments, we rotated the single-crystal samples by a given angle to probe the desired orientation of the crystal (see the main text). The dynamic loading was imposed from the rear side (the detector side) of the target via laser ablation with focused 800 nm laser pulses with pulse widths of 20 ps (FWHM) and pulse energies ≤10 mJ. The laser pulses were focused with an achromatic lens (focal length = 60 cm) onto the target with Gaussian-like intensity profiles (FWHM ~ 360 μm) (cf. Supplementary Fig. 1). Spatial overlap of the electron probe and optical pump was achieved with the fluorescence (and scattered light) from a thin YAG screen placed at the target position. Within the probed area (see Supplementary Fig. 1), the energy fraction is measured to be ~21%, yielding a maximum average fluence of ~4.3 ± 0.4 J cm$^{-2}$, with the uncertainty given by the rms intensity variation.

Temporal overlap between the optical pump and the electron probe was achieved using the Debye–Waller effect[29,30] on the Laue peak intensity decay of a laser-excited gold thin film (polycrystalline, 20-nm-thick) that was placed on the target plane. In this time zero measurement, the optical pump fluence was set at a value well below the damage threshold of gold thin film and the stroboscopic pump-probe measurement was performed at a repetition rate of 360 Hz for high data quality. The results are shown in Supplementary Fig. 2 illustrating the time history of the intensity of Laue peak (533). Here we define the time zero as the onset of the intensity decay, which corresponds to the overlap of the leading edges of the pump and the probe. For the compression experiments, we repeated the pump-probe measurement five times at each delay point, together with reference diffraction patterns of the same target obtained without the optical pump. Each

measurement was performed on a fresh sample. Diffraction patterns from these five target shots were analyzed individually, and the results of the Laue peak dynamics were averaged and shown in Supplementary Figs. 6 and 7.

MeV electrons offer unique diffraction characteristics, in particular the formation of a nearly flat Ewald sphere in the reciprocal lattice space. The formation of a "flat" Ewald sphere holds true for MeV electrons because their wave vector ($\sim 2 \times 10^{13}$ rad m$^{-1}$ for 3.7 MeV), which takes the radius of curvature of the Ewald sphere, is orders of magnitude higher than their scattering vector, e.g. $\sim 4 \times 10^{10}$ rad m$^{-1}$ for the {220} peaks. Therefore, the Ewald sphere can be treated as a plane on the finite region of the reciprocal space subtended by the detection area. This advantage allows us to detect simultaneously many diffraction orders of Laue peaks from the single-crystal aluminum.

**Sample preparation.** Samples were prepared by e-beam evaporation of a 200 ± 20 nm layer of Al onto optically polished (110)-oriented single-crystal NaCl flats (25 mm × 25 mm) at elevated temperatures. These single-crystal NaCl flats were purchased from Goodfellow Ltd. (https://www.goodfellow.com/) and have a nominal precision of ±3°. The fabricated samples were mosaic single crystals with highly oriented micrometer-sized grains[31]. The mass density of such targets is expected to be within 3% of the bulk density[32]. The deposited thin films were lifted off from the substrate in distilled water and transferred to the sample card. The sample card is a 2″ by 1″ silicon wafer that is etched with a 45 × 20 array of square windows of 500 μm × 500 μm in size.

**Molecular dynamics simulations.** We performed molecular dynamics (MD) simulations of the transverse and longitudinal elastic strain evolutions in single-crystal Al undergoing laser-driven dynamic compression. Given that a comprehensive full-scale simulation is challenging, we adopted a two-step simulation, shown in Supplementary Fig. 10, to provide a physical understanding of the key experimental results. This simulation method has been demonstrated in modeling the dynamic compression driven by picosecond laser ablation[33].

The first step is the one-dimensional (1D) hydrodynamics simulation to model the laser interaction with a solid sample. This was done with MULTI 1D code[34]. The simulation setup is the same as the experiment: a 200-nm-thick Al slab irradiated by a 800 nm, 20 ps (FWHM) laser pulse. We performed the simulations for three incident peak intensities of 0.06, 0.11, and 0.215 TW cm$^{-2}$ that correspond to the experimental pump fluences of 1.2, 2.2, and 4.3 J cm$^{-2}$, respectively. The absorption of the 800 nm light by Al in the simulations is found to be ~40% at the maximum peak intensity. The "shock" breakout time is found at ~32 ps and the maximum achievable stress is ~25 GPa as found in the case of 4.3 J cm$^{-2}$ in hydrodynamics simulations.

The second step is the MD simulation of the dynamic compression of the sample. This was done with the large-scale atomic/molecular massively parallel simulator (LAMMPS) code[22] using an embedded-atom method (EAM) potential for Al from Liu et al.[35]. The applicability of this EAM potential in dynamic compression conditions has been validated against experimental shock Hugoniot data[36]. In our MD simulations, Al single crystals are constructed with dimensions of about 200 × 100 × 100 nm$^3$, corresponding to a total of ~120 million atoms. Prior to the ramp loading, all the configurations are relaxed at 0 K, and then annealed with the constant-pressure–temperature ensemble under 3D periodic boundary conditions at 300 K and zero pressure. The dynamic compression simulations are performed with the microcanonical ensemble. The loading is applied along the x-axis, i.e. the [110] orientation of the sample. Periodic boundary conditions are applied along the y- and z-axes in the simulations, while a free boundary is applied along the loading direction. The time step for integration of the equation of motion is 1 fs.

The 20-nm-thick region on the left-hand side of the simulation box is set as the piston region to drive the compression wave. This 20-nm depth is comparable with the skin depth of the 800 nm light in Al. Further analysis on the hydrodynamics simulations shows that laser ablation process will cause an additional 7% in mass loss within a time window of 70 ps (Materials and methods are available as supplementary materials). This additional mass loss however will not significantly affect the elastic strain results that are shown in Fig. 3. The interactions between the piston and the rest of the atoms in the configuration are described with the same interatomic potential, while the atoms in the piston do not participate in molecular dynamics. The piston velocity history is determined from the MULTI 1D hydrodynamics simulations. Here we used the peak velocities before 35 ps (approximately at the shock breakout time) from the hydrodynamics simulations as the velocity history of the piston (see Supplementary Fig. 10). After 35 ps, the rarefaction waves will enter the sample. Further hydrodynamic simulations with the first 35 ps of the same optical pulse and the whole optical pulse showed that the residual pulse (>35 ps) has little influence on the compression due to the expansion of the plasma on the front surface and its absorption of the residual optical light. Supplementary Fig. 14 shows the temporal evolution of the pressure wave and velocity obtained from MD simulations for the three pump fluences employed in the UED experiment.

With the atomic configurations obtained from the MD simulations, we can calculate the longitudinal and transverse elastic strains. To compare with the UED results, we averaged the lattice strains among all the planes perpendicular to and parallel with the loading axis when computing the longitudinal elastic strain and

transverse elastic strain, respectively. For instance, for the loading along the [110] direction, the (220) lattice plane is selected for the longitudinal strain calculation, while the $(2\bar{2}0)$ plane, for the transverse elastic strain calculation. The simulated strain evolutions for the three pump fluences are presented in Fig. 3 of the main text. In addition, we can obtain von Mises shear strain and dislocation structure[37] at different time delays from the atomic configurations and investigate the relationship between plastic strain rate and dislocation motion, as shown in Fig. 4 in the main text. 1D binning analysis is applied to resolve spatial stress ($\sigma_{ij}$) and we define $2\tau = \sigma_{xx} - \frac{1}{2}(\sigma_{yy} + \sigma_{zz})$, where $\tau$ denotes shear stress.

## Data availability
The authors declare that the data supporting the findings of this study are available within the paper and its Supplementary Information file. All other relevant data supporting the findings of this study are available on request.

## Code availability
All software used in this study is freely available and can be accessed as follows: LAMMPS (https://www.lammps.org), Multi-1D (https://elsevier.digitalcommonsdata.com/datasets/zdcrnjph48/1).

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

## Acknowledgements

M.Z.M. thanks E. McBride, L. Fletcher, D. Swift, and J. Hasting for fruitful discussions and useful comments. We thank SLAC management for the strong support. We thank the technical support on single crystal film sample synthesis and characterization from the Center for Integrated Nanotechnologies (CINT), an Office of Science User Facility operated for the U.S. Department of Energy (DOE) Office of Science. This work was supported by the U.S. Department of Energy Contract No. DE-AC02-76SF00515 and the DOE Fusion Energy Sciences under FWP #100182. The experimental part of this research was performed at SLAC MeV-UED, which is supported in part by the U.S. Department of Energy, Office of Science, Office of Basic Energy Sciences (DOE BES) SUF Division Accelerator & Detector R&D program, the LCLS facility, and SLAC under contract nos. DE-AC02-05-CH11231 and DE-AC02-76SF00515. This work was performed, in part, at the Center for Integrated Nanotechnologies, an Office of Science User Facility operated for the U.S. Department of Energy (DOE) Office of Science. Los Alamos National Laboratory, an affirmative action equal opportunity employer, is managed by Triad National Security, LLC for the U.S. Department of Energy's NNSA, under contract 89233218CNA000001. M.Z.M. acknowledges the support from Department of Energy, Laboratory Directed Research and Development program at SLAC, under contract DE-AC02-76SF00515. M.X.T. and S.-N.L. acknowledge the support from National Natural Science Foundation of China under Grant No. 11627901.

## Author contributions

M.Z.M., X.J.W. and S.G. conceived and designed the study. M.Z.M., M.X.T., Z.J.C, J.R.P., X.Z.S., M.F., M.K., A.R., J.C.E, A.D., B.K.O., R.K.L., X.J.W. and S.G. performed the UED experiment, J.K.B. and Y.Q.W. fabricated the samples, M.Z.M. and J.R.P. analyzed the UED data, M.X.T., J.C.E and S.-N.L. performed and analyzed the molecular dynamics and UED simulations, M.Z.M., M.X.T., S.-N.L. and S.G. wrote the manuscript with revisions from all authors.

## Competing interests

The authors declare no competing interests.
