## [Peer Review File · Nature Communications]

Title: Ultrafast visualization of incipient plasticity in dynamically compressed matterREVIEWER COMMENTS

Reviewer #1 (Remarks to the Author):

The paper, “Ultrafast visualization of incipient plasticity in dynamically compressed matter” describes a beautiful experiment to study elastic to plastic transition with sufficient time resolution to observe incipient plasticity. The inclusion of supporting molecular dynamic simulations with good agreement is encouraging. The paper definitely needs to be published but needs a little work to improve clarity. The electron beam size is given in the experimental section, but should be provided earlier; because the authors describe the atomic level of dislocation dynamics a reader might be left with the impression that they are using an angstrom level probe, yet useable momentum transfer resolution requires coherent superposition of scattering from millions of atoms. I think this is further confused by the authors use of 1D and 3D when talking about reciprocal space as opposed to real space. However the sample is macroscopically 2D in nature with an underlying 3D crystal anisotropy probed by a macroscopic local 1D pulse so I’m a little confused on what they are trying to say. Another confusing comment in the paper has to do with an increase in the intensity of peaks due to a realignment of grains. Is this a perfect single crystal, a mosaic single crystal or a polycrystal? The authors talk about a single shot experiment, but 5 pulses per shot. I can’t figure out if the detector can resolve each electron pulse or if there is an averaging of the pattern over 5 pulses using a gating mechanism on the detector? Also, the terse description of the temporal overlap measurement needs probably one more sentence to make it clear. How exactly was this done? Was the powder pattern generated by the electron beam which overlapped the sample and the monitor polycrystal? Was the same detector used to collect the Al Laue pattern and the Au powder pattern? Fix these problems and the paper will be an important addition to the literature.

Reviewer #2 (Remarks to the Author):

The manuscript describes results of combined experimental and hydrodynamics/molecular dynamics (MD) studies of incipient plasticity in single crystal Al under ultrafast dynamic compression. The dynamics of materials response is studied using fs MeV electron diffraction. By analyzing temporal evolution of Laue diffraction peaks for {220} family of planes in Al, the authors determined the temporal evolution of longitudinal and transverse elastic strains and then using Orowan’s equation derive the mobile dislocation density, which then is compared with that from MD. The authors claim a perfect agreement between experiment and MD, which justifies extraction of dislocation dynamics from MD.

Although the results are interesting and potentially important, current description of experimental conditions, analysis of experimental results, hydrodynamic and MD simulations is incomplete. In particular a serious issue with analysis of Laue diffraction of dynamically compressed Al is a neglect of the effect of crystal rotation which was shown is equally important as plastic deformations due to shear under uniaxial compression, see, e.g. Zaretsky (JAP, 93, 2496 (2003)). Another outstanding problem – a disregard a contribution of stacking faults to diffraction. In particular, using simulation results from MD,

the authors make a conclusion of a substantial fraction of Shockley partials (Fig. 3D). If this is the case, then there should be a large number of stacking faults with end result of shifts of the diffraction peaks in opposite direction than dislocation-mediated plasticity. This effect is fully described in a classic textbook “X-ray Diffraction” by Eugene Warren.

The most serious problem with the manuscript is that the authors do not provide any comprehensive description of the state of the crystal irradiated by femtosecond laser pulse experiencing dynamic compression of the Al single crystal. What is the character of this compression, ramp, shock, velocity of the wave, pressure profile, its time evolution? These are the major pieces of information that are critically important for specifying the actual physical state of compression. Without clear understanding the driving force of the dynamic compression, discussion of fine details of dislocation dynamics does not make sense. It is not clear whether the authors have attempted to acquire velocimetry data to fully characterize the compression wave. It is not simple in case of dynamic compression taking place at ps time scale, but it has been done as shown by LANL (Shawn McGrain et al) and LLNL (Mike Armstrong et al) groups.

The description of the hydrodynamic model of laser ablation causing the dynamic compression is lacking. Using simplified coupling of the temporal development of stress from hydrodynamics with that in MD, is not justified. Driving compression in MD by fixed layer of atoms does not take into account a complex process of formation of the stress state in the ablation process including potential expansion of the ablated surface of the irradiated material. More insight into these critical steps is needed.

The MD simulations play a key role in interpreting experimental results, but its validation is not provided. The quality of interatomic potentials plays a key role in delivering trustworthy MD results. The EAM potential by Wang et al. used in MD simulations was developed for description of Al at ambient pressure. It is not clear whether compressions including plasticity under dynamic compression are adequately described. There are plenty of experimental shock Hugoniot data that can be easily used for validation of MD, and the authors should do a better job in this regard before claiming a perfect agreement between experiments and MD simulations. The authors should provide detailed description of MD results, including temporal evolution of the pressure wave, its velocity and other key information which is readily available from MD. It might not fit into the main text, but can be out in the Supplementary Information section.

Reviewer #3 (Remarks to the Author):

This manuscript reports femtosecond MeV diffraction measurements to quantify the shock compression induced elastic-plastic response. The strains measured from peak spacings are used to analyze the strains generated. MD simulations are used to understand the dislocation nucleation and evolution behavior to couple with experiments.

While there is a novelty in the ability to use in situ diffraction to analyze deformation response, there

are several aspects that raise questions about the interpretations of the mechanisms and understanding of plasticity mechanisms that justify publication in Nature Communications. A few other concerns:

1. The study uses 200 nm single-crystal Al [110] film samples prepared using e-beam evaporation. Is the microstructure truly single-crystal or a polycrystalline microstructure with the [110] texture? This is unclear as a statement on Page 6 states that the increase in intensities of LSPPs due to the re-alignment of grains. What is the grain size of the microstructure? Why are there no peaks in the diffractograms?
2. The shock loading is carried out via ablation using a 20 ps and 10 mJ laser. The interaction of Al with a laser that involves ablation is likely to result in the melting of the metal before the generation of the shock wave. In addition, temperatures generated are significantly higher in the shock front. There is no discussion of laser melting and the role of temperatures on the transverse and longitudinal strains.
3. The strains are evaluated based on the distances between Laue peaks. An elastic-plastic transition is suggested at a fluence of 4.3 J/cm² and the co-existence of the longitudinal and transverse strains is identified as an indicator of plasticity and a two-wave structure of the shock wave. However, the understanding of a two-wave structure is very clear in the literature.
4. The inference of dislocation densities based on strains is observed to peak during stage II. The prediction of dislocation densities based on strains does not account for the temperature-dependent aspects of plasticity and not based on the broadening of the peaks.
5. The experiments are complemented using MD simulations that use a 1D hydro-code to predict the ramp duration of the piston to mimic shock loading. Again, such a framework does not account for the ablation/melting behavior of the metal before a shock wave is generated. Such effects are related to the absorption of laser energy by electrons and then transfer to the lattice through electron-phonon coupling. Such an MD framework does not mimic the laser loading conditions and the onset of plasticity at 27 ps is an artificial correlation with experiments. The experiments only discuss mobile densities and no correlation with the distribution of The suggestion of the creation of dislocation embryos is not clear.
6. The sample dimensions of 200 nm are very small to get enough insights into the elastic-plastic transitions during laser-shock loading. The phenomena of ablation and melting are likely to consume a good fraction of the metal during the generation of the shock wave. Why are the effects of the solid-liquid interface in the microstructure not observed in the diffractograms?
7. The generation of plasticity through dislocations is likely to create a high density of stacking faults in the microstructure. Would these stacking faults with an hcp structure provide additional peaks in the diffractograms?

There are many more challenges in the interpretation of the results discussed in this manuscript given the small dimensions of the Al sample, generation of the shock using a laser, and the interpretation of dislocation densities based only on strains using diffractograms. The experimental data and the modeling data are two disconnected discussions in the manuscript.

Nonetheless, there are no new insights that improve our understanding of the deformation response of fcc metals. The manuscript is therefore not recommended for publication.

REVIEWER COMMENTS

Response to Reviewer 1:

We acknowledge the reviewer for the valuable comments and questions that helped us to improve our manuscript. We also thank the reviewer for recognizing the significance of our results, in particular the direct observation of incipient plasticity. We have addressed all the questions and complied with the suggested changes from the reviewer. Our detailed responses are found as follows.

Reviewer #1 (Remarks to the Author):

From reviewer's report:

The paper, "Ultrafast visualization of incipient plasticity in dynamically compressed matter" describes a beautiful experiment to study elastic to plastic transition with sufficient time resolution to observe incipient plasticity. The inclusion of supporting molecular dynamic simulations with good agreement is encouraging. The paper definitely needs to be published but needs a little work to improve clarity.

Response: we thank the reviewer for the thorough review of the manuscript and for the encouraging comment. We have addressed all the concerns and the following are the detailed responses.

From reviewer's report:

1. The electron beam size is given in the experimental section, but should be provided earlier; because the authors describe the atomic level of dislocation dynamics a reader might be left with the impression that they are using an angstrom level probe, yet useable momentum transfer resolution requires coherent superposition of scattering from millions of atoms.

Response: we comply with the reviewer and add the information of the electron beam size in the last sentence of the second paragraph on Page 3. This sentence now reads:

“In the meantime, a 600-fs-long bunch length of the relativistic electrons (~ 220 μm in diameter) makes it possible to probe the ultrafast microstructural evolution during plastic deformation.”

From reviewer's report:

2. I think this is further confused by the authors use of 1D and 3D when talking about reciprocal space as opposed to real space. However the sample is macroscopically 2D in nature with an underlying 3D crystal anisotropy probed by a macroscopic local 1D pulse so I'm a little confused on what they are trying to say.

Response: We would like to clarify the use of 1D and 3D in the discussion of reciprocal space and real space. In our experiment, the sample was rotated by an angle away from the normal incidence such that the e-beam propagation axis is in slanted angle with the a, b, c axes of the unit lattice (Figure 1 B of the main text). In this way, any dimension of the lattice motion during the dynamic compression will have a projection orthogonal to the e-beam axis and therefore can be detected with the peak shifts of corresponding diffraction spots. More specifically, the peak shifts of $(2\bar{2}0)$ and $(\bar{2}20)$ peaks [Figure 1A of the main text] will reflect the lattice deformation along the y-direction (in-plane axis) and the other four peaks will reflect the deformation along the loading direction (out-of-plane axis) and the other transverse direction (in-plane axis). For more information on 3D structural dynamics studies with MeV-UED, please refer to the work by Reid et al. [A. Reid, et al. Beyond a phenomenological description of magnetostriction, Nat. Commun. 9, 388 (2018)].

From reviewer's report:

3. Another confusing comment in the paper has to do with an increase in the intensity of peaks due to a realignment of grains. Is this a perfect single crystal, a mosaic single crystal or a polycrystal?

Response:

Our thin film target is a mosaic single crystal. The mosaicity is mainly caused by the NaCl single-crystal substrates (for sample deposition) which have misorientation of $\pm 3^\circ$ (Sample Preparation Section in the supplementary information). Our hypothesis on the increase in the intensity of peaks is illustrated in Figure R1. The mosaic single crystal is made up of small grains/blocks with varying orientations. The laser ablation pressure can reduce the mis-orientation of the grains, resulting in an increase of the Laue diffraction peak intensities.

Figure R1 Schematic diagram illustrating the high pressure-induced grain alignment of a mosaic single crystal.

From reviewer's report:

4. The authors talk about a single shot experiment, but 5 pulses per shot. I can't figure out if the detector can resolve each electron pulse or if there is an averaging of the pattern over 5 pulses using a gating mechanism on the detector?

Response:

We thank the reviewer for pointing out this confusing point. Our experiment was performed in a single-shot mode with the detector resolving each diffraction pattern. In the Experimental Details Section, we stated that “*For the compression experiments, we acquired five pump-probe shots for data averaging at each time point, together with respective ambient diffraction patterns of the same target when the optical pump is switched off*”. What we mean here is that at each delay time, we repeated the pump-probe diffraction measurement five times; the five pump-probe patterns were analyzed individually, and the results of the Laue peak dynamics were averaged; the experimental data shown in Supplementary Figure 6 (peak shifts of {220} spots) and Supplementary Figure 7 (intensities and widths of {220} spots) are the averaged results of the five pump-probe shots. To comply with the reviewer, we modified this sentence to the following (the last sentence of the first paragraph on page 13.) for better clarity:

“*For the compression experiments, we repeated the pump-probe measurement five times at each delay point, together with reference diffraction patterns of the same target obtained without the optical pump. Each measurement was performed on a fresh sample. Diffraction patterns from these five target shots were analyzed individually, and the results of the Laue peak dynamics were averaged and shown in Supplementary Figure 6 and Supplementary Figure 7.*”

From reviewer's report:

5. Also, the terse description of the temporal overlap measurement needs probably one more sentence to make it clear. How exactly was this done? Was the powder pattern generated by the electron beam which overlapped the sample and the monitor polycrystal? Was the same detector used to collect the Al Laue pattern and the Au powder pattern? Fix these problems and the paper will be an important addition to the literature.

Response:

We thank the reviewer for recognizing the significance of this work. For the temporal overlap measurement, we employed a widely used method called Debye-Waller effect to find the time zero between the optical pump and the electron probe (See references: Sokolowski-Titen et al. Struct. Dyn. 4, 054501 (2017), Mo, et al. Science 360, 1451 (2018) and references therein). In this scheme, the thermal heating from the optical excitation results in an increasing vibration of atoms, which leads to a decay of the intensity of the Laue diffraction peaks, as seen in Supplementary Figure 2 in the Supplementary Information.

In our time zero measurement, we utilized the same experimental setup and geometry as the dynamic compression experiments, except the following changes: (a) a polycrystalline gold nanofoil, mounted on the same target plane as the aluminum samples, was used as the target since it has a higher damage threshold and its DW effect is well understood from the past UED experiments (Sokolowski-Titen2017 and Mo2018); (b) the optical fluence was controlled well below the damage threshold of gold and (c) the stroboscopic pump-probe measurement was run at a repetition rate of 360 Hz for data quality. Once the time zero was determined, we switched back to the single shot mode for the dynamic compression experiment of aluminum. To answer the remaining questions from the reviewer: yes, the powder pattern was generated from the electron beam which overlapped with the sample and the monitor polycrystal; the same detector was used between time zero measurement and the Al dynamic compression experiment.

To comply with the reviewer, we added the references for Debye-Waller effect described in the first sentence of Paragraph 1 on Page 13, and added the following sentence (highlighted by blue) after that sentence:

“Temporal overlap between the optical pump and the electron probe was achieved using the Debye-Waller effect (Sokolowski-Titen2017 and Mo2018) on the Laue peak intensity decay of a laser-excited gold thin film (polycrystalline, 20-nm-thick) that was placed on the target plane. In this time zero measurement, the optical pump fluence was set at a value well below the damage threshold of gold thin film and the stroboscopic pump-probe measurement was performed at a repetition rate of 360 Hz for high data quality.”

Response to Reviewer 2

We thank the reviewer for the valuable comments that helped us to improve the quality of our manuscript. We have addressed all the comments from the reviewer and modified our manuscript accordingly. Our detailed responses are found as follows.

Reviewer #2 (Remarks to the Author):

From reviewer's report:

The manuscript describes results of combined experimental and hydrodynamics/molecular dynamics (MD) studies of incipient plasticity in single crystal Al under ultrafast dynamic compression. The dynamics of materials response is studied using fs MeV electron diffraction. By analyzing temporal evolution of Laue diffraction peaks for {220} family of planes in Al, the authors determined the temporal evolution of longitudinal and transverse elastic strains and then using Orowan's equation derive the mobile dislocation density, which then is compared with that from MD. The authors claim a perfect agreement between experiment and MD, which justifies extraction of dislocation dynamics from MD.

1. Although the results are interesting and potentially important, current description of experimental conditions, analysis of experimental results, hydrodynamic and MD simulations is incomplete. In particular a serious issue with analysis of Laue diffraction of dynamically compressed Al is a neglect of the effect of crystal rotation which was shown is equally important as plastic deformations due to shear under uniaxial compression, see, e.g. Zaretsky (JAP, 93, 2496 (2003)).

Response: we thank the reviewer for acknowledging the importance of our work. We complied with the reviewer and added in the revised manuscript the suggested details for the discussion of experiments and simulations, which are also provided in our responses to the reviewer's comments.

We agree with the reviewer that crystal rotation is important in plastic deformation under uniaxial compression. In the experiment by Zaretsky et al. (JAP2003), multiple diffraction peaks were captured from the shocked NaCl single crystal with the (200)-peak shift depending on the longitudinal elastic strain and the (220)-peak depending on both longitudinal and transverse elastic strains. Due to the special geometry in which the (200) lattice plane was orthogonal to the loading direction, crystal rotation during plastic deformation did not affect the peak shift of the (200) peak. In contrast, the (220) peak shift was subject to crystal rotation. Therefore, to solve for the transverse elastic strain, the crystal rotation effect needed to be considered. However, near the hydrostatic condition, the crystal rotation effect was found to be completely canceled out by the shear strain, resulting in a relative peak shift of (220) equal to the transverse elastic strain.

For our UED experiment, we also captured multiple diffraction peaks for measuring the longitudinal and transverse elastic strains. Differing from Zaretsky's geometry, we have two diffraction peaks, i.e. the two peaks labelled by TSPP [(2-20) and (-220)], that probe the transverse lattice compression. Figure R2 shows the schematic for the dominant slip system and crystal rotation with respect to the diffraction planes of TSPP. The loading is applied along the sample normal direction, i.e. [110], and the slip plane (111) is normal to the diffraction planes of TSPP. The crystal rotation takes place about the axis that is normal to the indicated slip direction, which in our case leads to a crystal rotation in the diffraction planes of TSPP. The crystal rotation in this way will not alter the spacing and

orientation of TSPP diffraction planes [Wehrenberg et al, Nature 550, 496 (2017)], hence it will not affect the peak shift of TSPP. Therefore, the peak shift of TSPP depends only on the transverse elastic strain. Similar method of directly measuring transverse elastic strain is also found elsewhere [Whitlock et al, Phys. Rev. B 52, 1 (1995)]. With this and Taylor's theory stating that the total transverse strain equals to zero in uniaxial compression, i.e., $\epsilon_t = \epsilon_t^e + \epsilon_t^p = 0$, one obtains the relationship of $\epsilon_t^p = -\epsilon_t^e$. This implies that our measured transverse elastic strain from TSPP has the same absolute magnitude as the transverse plastic strain.

Figure R2 Schematic showing the dominant slip system and crystal rotation with respect to the diffraction plane (2-20) of TSPP during the plastic deformation in (110)-oriented single-crystal Al.

To resolve the concern on crystal rotation, we added the following sentences in the “Transverse elastic strain” subsection (Page 5) of Supplementary Information:

“Our pump-probe geometry directly measures the crystal compression normal to the loading direction via TSPP [(2-20) and (-220)]. Under our experimental condition with the loading along the sample normal direction, i.e. [110], the {111} slip planes pertinent to plastic deformation are normal to the diffraction planes of TSPP. The resulted crystal rotation occurs in these diffraction planes and will not affect the peak shift of the TSPP [Wehrenberg et al, Nature 550, 496 (2017)]. Therefore, the peak shift of TSPP depends only on the transverse elastic strain, ϵ_t^e , which can be computed using the following equation :...”

From reviewer's report:

2. Another outstanding problem – a disregard a contribution of stacking faults to diffraction. In particular, using simulation results from MD, the authors make a conclusion of a substantial fraction of Shockley partials (Fig. 3D). If this is the case, then there should be a large number of stacking faults with end result of shifts of the diffraction peaks in opposite direction than dislocation-mediated plasticity. This effect is fully described in a classic textbook “X-ray Diffraction” by Eugene Warren.

Response: We thank the reviewer for pointing out the stacking fault issue. We complied with the reviewer and evaluated the effect of stacking faults on the diffraction peak shifts.

We start by noting that, for FCC metals containing stacking faults, some (hkl) peaks from a given {hkl} family are shifted by the stacking faults (affected components) while others are not shifted (unaffected components). The affected components are characterized by $|L_0| \equiv |h + k + l| \equiv 3J \pm 1$ (J is an integer) whereas the unaffected ones follow $|L_0| \equiv 3J$ (J is an integer). The above laws are described in Warren's textbook and other references such as the work by Sharma et.al. [Sharma et al., PRX 10, 11010 (2020)]. In our case, the {220}-family peaks follow $|L_0| = 0$, implying that their peak shifts are not affected by the stacking faults.

To confirm this, we performed diffraction simulations of the MD atomic trajectories, from which the effect of stacking faults on the peak shifts can be studied [A. Mishra, *et. al.*, Sci. Rep. 11, 9872 (2021)]. The electron diffraction simulations were conducted using the code GAPD (GPU-accelerated Atom-based Polychromatic Diffraction) developed by Luo's group (J. C. E, *et. al.*, J. Synchrotron Rad. 25, 604 (2018)). Figure R3 shows the intensity lineout of the simulated (220) spots of the MD atomic trajectory obtained at 42 ps for the case with pump fluence of 4.3 J cm⁻². Following the same XRD analysis by Mishra et al., we generated diffraction intensity results for full structure (orange line), FCC atoms only (green line), and the stacking faults atoms (black line). Note that there are about 8.9% stacking faults contained in the full structure at 42 ps. However, as shown in the Figure R3, there is hardly any peak shift that can be identified by comparing the full structure result with FCC-only result. This means that the effect from the stacking faults on the peak shift of our measured {220} peaks is negligible.

Figure R3: Intensity lineout of the simulated (220) diffraction spot for full microstructure (orange line), FCC only atoms (green line), and HCP atoms of the stacking faults (black line) of the shocked Al at delay time of 42 ps for the pump fluence of 4.3 J cm^{-2} .

To resolve the concern on stacking faults, we attached the following sentences in the first paragraph of Page 8 in the revised manuscript:

“MD simulation also reveals the presence of stacking faults with the generation of Schockley partial dislocations. For example, at $t = 42 \text{ ps}$, approximately 8.9% of stacking faults is found in the full simulated structure for the highest fluence case. The effect of these stacking faults on diffraction peak shifts is evaluated and the result indicates that stacking faults will not affect the {220}-family peaks employed to infer elastic strains in this study.¹⁹” [Ref 19: Supplementary information]

Furthermore, we added a section named “Effect of stacking faults on diffraction peaks” (Page 10) in the Supplementary Information of our revised manuscript.

From reviewer's report:

3. The most serious problem with the manuscript is that the authors do not provide any comprehensive description of the state of the crystal irradiated by femtosecond laser pulse experiencing dynamic compression of the Al single crystal. What is the character of this compression, ramp, shock, velocity of the wave, pressure profile, its time evolution? These are the major pieces of information that are critically important for specifying the actual physical state of compression. Without clear understanding the driving force of the dynamic compression, discussion of fine details of dislocation dynamics does not make sense. It is not clear whether the authors have attempted to acquire velocimetry data to fully characterize the compression wave. It is not simple in case of dynamic compression taking place at ps time scale, but it has been done as shown by LANL (Shawn McGrain et al) and LLNL (Mike Armstrong et al) groups.

Response: We agree with the reviewer that the knowledge of the compression wave is important for dynamic compression studies, and we thank the reviewer for suggesting the velocimetry approach to characterize the compression wave. However, we don't think the velocimetry approaches by Shawn McGrain and Mike Armstrong would be able to fully characterize the compression wave in our case. This is because the shock wave generated in our experiment is not fully supported due to the fact the laser pulse length (20 ps) is shorter than the transit time (32 ps) for the sound wave through the sample. As such, we will need resort to hydrodynamics simulations to understand the compression wave, as has been done in previous LCLS studies [D. Milathianaki, et al. Science 342, 220 (2013) and J. Coakley et al. Science Advances 6, eabb4434 (2020)].

To comply with the reviewer, we included the time history of peak particle velocity, and the corresponding pressure profile for the highest fluence of 4.3 J cm^{-2} as an insert in Figure 1 of the main manuscript. For the ease of your review, we present the inert here as Figure R4.

Figure R4: MULTI-1D simulated time history of the peak particle velocity and peak pressure in a 200nm-thick Al irradiated with a laser fluence of 4.3 J cm^{-2} . This figure is added as an insert in Figure 1 of the main manuscript.

From reviewer's report:

4. The description of the hydrodynamic model of laser ablation causing the dynamic compression is lacking. Using simplified coupling of the temporal development of stress from hydrodynamics with that in MD, is not justified. Driving compression in MD by fixed layer of atoms does not take into account a complex process of formation of the stress state in the ablation process including potential expansion of the ablated surface of the irradiated material. More insight into these critical steps is needed.

Response: We are happy to provide more information to justify the two-step simulation methodology for modelling dynamic compression induced by laser ablation.

First, such a two-step simulation methodology has been demonstrated in modelling the dynamic compression driven by picosecond laser ablation, see the studies led by Milathianaki et al. [1-2]. In these two studies, 1- μm thick polycrystalline copper films were strained by picosecond laser ablation and probed *in-situ* with femtosecond XFEL pulses. For the simulation, hydrodynamics simulation using the HYADES code [3] was first employed to model the laser-plasma interaction. The calculated stress history was then incorporated into the Dislocation Dynamics [1] or Molecular Dynamics (LAMMPS) [2] to model the deformation dynamics of the compressed matter. In both cases, the hydro stress profile was imposed on the front surface of the target which then acted as a variable force piston to compress the sample. Likewise, the first 20-nm layer of the sample in our MD simulations was selected as the piston region to drive the compression with the velocity profile obtained from hydrodynamics simulation. This thickness is sufficient to account for the mass loss due to the ablation process, which is supported by our hydrodynamics simulations (See our response to the 2nd part of this comment) and the work published by Coakley et al. [2].

Second, we were able to reproduce the three data sets of elastic strains obtained at different pump fluences using this two-step simulation approach. The lowest fluence data can serve as a good benchmark for the coupling of the two simulation codes since the effect of laser ablation process is the minimum for the dynamic loading. More importantly, the simulations predicted the right fluence threshold for elastic-plastic transition, i.e., the 2.2 J cm⁻². At this fluence condition, the pressure is slightly lower than the elastic limit and hence no plasticity is observed in the MD simulation, agreeing with the UED experiment.

References:

- [1] D. Milanthianaki, et al. Science 342, 220 (2013).
- [2] J. Coakley et al. Science Advances 6, eabb4434 (2020).
- [3] J. T. Larsen, et al. J. Quant. Spectrosc. Radiat. Transfer 51, 179 (1994).

With respect to the second part of the comment: “*Driving compression in MD by fixed layer of atoms does not take into account a complex process of formation of the stress state in the ablation process including potential expansion of the ablated surface of the irradiated material. More insight into these critical steps is needed*”, we thank the reviewer for granting us the opportunity to explain the effect of the ablation process.

In what follows, we provide additional details on hydrodynamics simulations and discuss how the ablation process affects the dynamic compression process. The new analysis shows that the laser ablation process will cause an additional 7% in mass loss within a time window of 70 ps. This

additional mass loss will not significantly affect the elastic strain results that are shown in Figure 3 of the main text.

Figure R5 MULTI-1D simulation results of the temporal evolution of the mass density distribution for a 200-nm thick Al irradiated by 20-ps (FWHM), 800nm laser pulses at incident fluences of 1.4 J cm⁻² (A1), 2.2 J cm⁻² (B1) and 4.3 J cm⁻² (C1). The laser impinges on the target at x = 0 nm. The same color axis is applied to all the three false-color images with the representative color bar shown in (C1). The vertical dashed lines (x = -20 nm) mark the depth of 20nm from the target surface. (A2) to (C2) plot the temporal evolution of the fraction of the mass contained in the ROI with x ≥ -20 nm for the three respective pump fluences.

Figure R5 (A1) - (C1) shows the MULTI-1D simulated mass density as a function of time and space for a 200 nm thick Al irradiated by 20 ps (FWHM), 800 nm laser pulses at three pump fluences of 1.2 J cm⁻², 2.2 J cm⁻² and 4.3 J cm⁻², respectively. The simulation results imply a clear dependence of the ablation process on the incident pump fluence. The onset time of the front surface expansion starts at ~20 ps for 1.2 J cm⁻² and is advanced to ~10 ps for 4.3 J cm⁻², which leads to the differences in the shock wave propagation and the breakout at the rear surface.

To quantify the effect from the expansion at the front surface, we track the change of mass contained in the region of interest (ROI) with x ≥ -20 nm. We divided the mass of the ROI at each time delay by the total mass, yielding the time evolution of the mass fraction of ROI, as shown by the black lines in Fig. R5 (A2) to (C2). Similar trend is observed for the three fluences: the mass fraction is maintained the same before the onset of hydrodynamic expansion and then increases by ~7% by the end of 70 ps. Increasing the incident fluence will advance the onset time for expansion as well as accelerating the increase of the mass fraction with time. The additional 7% of the mass loss due to the plasma expansion, which is less than the 10% uncertainty of our sample thickness, will not

significantly affect the simulated elastic strain results that are shown in Figure 3 of the main text. For instance, MD simulations using a 35-nm front layer (~ 17 % thickness) as the piston region leads to ϵ_l^e of 0.081 at 35 ps, as comparing to the counterpart of 0.076 for the case with 20 nm thickness. Furthermore, the mass density of the ablated plasma is orders of magnitude lower than the solid density, as shown in Figure R5 (A1) - (C1). This indicates that the ablated mass will form a plasma gas cloud in front of the compressed sample and the electron scattering signal from the plasma gas will lead to an increase of the overall background signal.

To resolve the concerns from this comment, we made three changes in the revised manuscript and the supplementary information.

1. Added the following sentence in the first paragraph of “Molecular Dynamics Simulation” (Page 14):

“These simulations were based on a two-step simulation method, shown in Supplementary Figure 10, which has been demonstrated in modelling the dynamic compression driven by picosecond laser ablation³⁴.” (Ref 34: J. Coakley et al. Science Advances 6, eabb4434 (2020))

2. On Page 15, we added the following statement (colored by blue):

“...This 20-nm depth is comparable with the skin depth of the 800 nm light in Al. Further analysis on the hydrodynamics simulations shows that laser ablation process will cause an additional 7% in mass loss within a time window of 70 ps.¹⁹ This additional mass loss however will not significantly affect the elastic strain results that are shown in Figure 3. The interactions between the piston and the rest...”

3. In Supplementary Information, we added a section named “Hydrodynamic modelling of laser ablation” (Page 11).

From reviewer's report:

5. The MD simulations play a key role in interpreting experimental results, but its validation is not provided. The quality of interatomic potentials plays a key role in delivering trustworthy MD results. The EAM potential by Wang et al. used in MD simulations was developed for description of Al at ambient pressure. It is not clear whether compressions including plasticity under dynamic compression are adequately described. There are plenty of experimental shock Hugoniot data that can be easily used for validation of MD, and the authors should do a better job in this regard before claiming a perfect agreement between experiments and MD simulations. The authors should provide detailed description of MD results, including temporal evolution of the pressure wave, its velocity and other key information which is readily available from MD. It might not fit into the main text, but can be out in the Supplementary Information section.

Response: We agree with the reviewer that the choice of EAM potential plays a key role in MD simulations. Note that the EAM potential employed in our MD simulations was developed by Liu et al. [Liu et al., Modelling Simul. Mater. Sci. Eng. 12, 665 (2004)]. The applicability of this EAM potential in dynamic compression conditions has been assessed by Luo's group [Wang et al. JAP 117, 084301(2015)]. The results are shown in Figure R6. As indicated, MD simulations for shock wave loaded single-crystal Al at different orientations showed good agreement with existing Hugoniot data. We applied this EAM potential to model the plastic deformation and the results show surprisingly good agreement with the UED data.

To resolve the concern in this comment, we made two changes in the revised manuscript.

1. Page 15, we added a sentence on the applicability of EAM potential (colored by blue)

“...using an embedded-atom method (EAM) potential for Al from Liu et al.³⁶ The applicability of this EAM potential in dynamic compression conditions has been validated against experimental shock Hugoniot data.³⁷ In our MD simulations, ...” (Ref 37: Wang et al. JAP 117, 084301(2015))

2. We changed the wording for the agreement between simulation and experiment in the revised manuscript (three places):

Last sentence of abstract: Large-scale molecular dynamics simulations show good agreement with the experiment and provide an atomic-level description of the dislocation-mediated plasticity. [changed “excellent agreement” to “good agreement”]

End of introduction: MD simulations show good agreement with experiments and corroborate our observation of the dislocation-mediated plasticity. [changed “excellent agreement” to “good agreement”]

End of conclusion: The good agreement between experiments and simulations provides a full atomic-level picture of the ultrafast elastic-plastic deformation in single-crystal materials. [changed “remarkable agreement” to “good agreement”]

Figure R6 Validation of the Al EAM potential developed by Liu et al. [Ref. 32 of the main text] in dynamic compression conditions [Wang et al., JAP 117, 084301 (2015)]. (a) Shock velocity–particle velocity or u_s – u_p plots obtained from MD simulations and experiments (black dots), and (b) the corresponding plots of σ_{xx} vs. normalized specific volume (V/V_0). This figure is adopted from Figure 1 of the paper by Wang et al (JAP2015).

Furthermore, as suggested by the reviewer, we added the temporal evolutions of the pressure wave and velocity from MD simulations as Supplementary Figure 14 in the *Supplementary Information*. In the revised manuscript, we added the following sentence in Page 16 to connect with Supplementary Figure 14:

“Supplementary Figure 14 shows the temporal evolution of the pressure wave and velocity obtained from MD simulations for the three pump fluences employed in the UED experiment.”

For the ease of your review, we present Supplementary Figure 14 here as well, as indicated by Figure R7.

Figure R7: MD simulation results of the pressure (top row) and velocity (bottom row) profiles for a 200-nm thick Al irradiated at incident fluences of 1.4 J cm⁻² (A), 2.2 J cm⁻² (B) and 4.3 J cm⁻² (C).

Response to Reviewer 3

We sincerely appreciate the reviewer's evaluation and insightful comments to our manuscript. In the following, we would like to offer additional context and clarifying information on the novelty and impact of this work. Regarding the broader aspects of this work, we agree that the plastic deformation at high strain rates has been studied in the literature, which we acknowledged in the original manuscript. However, we believe that the novelty and significance of this work is the first experimental measurement of the incipient plasticity using the ultrafast MeV-UED instrument. Indeed, measuring these ultrafast structural dynamical processes at high strain rate conditions has been challenging due to the experimental limitations on spatial and temporal resolution. Our study on the lattice response of single-crystal aluminum to laser-induced ramp compression provides a complete time history of the incipient plasticity, enabling the determination of the dislocation nucleation and transport processes. So far, such quantitative measurements are not yet available in the literature. Apart from this scientific finding, another novel aspect of this work is the first deployment of MeV-UED in studying dynamic compression physics, which was also recognized by the reviewer. In particular, the flat Ewald sphere of MeV electrons permits to access a large momentum transfer range and many orders of diffraction spots from single-crystalline materials. This advantage was a critical factor to achieve our finding and will certainly be an important one for studying other high-pressure and high strain-rate phenomena such as melting line and phase transitions.

Again, we sincerely thank the reviewer for all the comments, and we hope that this detailed discussion is helpful to appreciate the novelty and broader impact of this work. We expect our work to be of interest to a broad audience in dynamic compression.

In the following, we provide specific responses to the questions and comments raised by the reviewer.

From reviewer's report:

1. The study uses 200 nm single-crystal Al [110] film samples prepared using e-beam evaporation. Is the microstructure truly single-crystal or a polycrystalline microstructure with the [110] texture? This is unclear as a statement on Page 6 states that the increase in intensities of LSPPs due to the re-alignment of grains. What is the grain size of the microstructure? Why are there no peaks in the diffractograms?

Response: Our samples are mosaic single crystals with highly oriented grains. The grain size is on the order of microns [see Grain size estimate for more details]. The static diffraction patterns of our samples show single-crystal diffraction instead of powder or textured rings.

To resolve this comment, we added the following sentence in the revised manuscript (Sample Preparation) to clarify the state of crystal.

“The fabricated samples were mosaic single crystals with highly oriented micrometer-sized grains³²”. (Ref 32: S. Sugawara, Materials Transactions, JIM, 1293, (1996))

Grain size estimate: we first estimate the average grain size L using the Scherrer equation [Warren X-Ray Diffraction (1990)]:

$$L = \frac{0.94\lambda}{B \cos\theta}$$

where $\lambda = 0.297$ nm is the de Broglie wavelength of 3.7 MeV electrons, B is the FWHM width of the diffraction peak, and θ is the diffraction angle of the peak. Here, we use the width of {111}-family peaks obtained at normal incidence to estimate the grain size. We obtain $L \sim 297$ nm using $\theta = 0.64$ mrad and $B = 0.94$ μ rad for the (111) peaks after deconvoluting the instrumental broadening function. Since Scherrer equation provides a lower bound of grain size, the obtained value implies that the grain size of our samples is greater than a few hundred nanometers.

On the other hand, the grain size of nanofilms has been studied extensively in the literature. Our thin film samples were fabricated by e-beam evaporation on [110] NaCl substrates that were heated to a temperature of 400°C. This is a well-known technique for epitaxially growing aluminum nano-films [S. Sugawara, Materials Transactions, JIM, 1293, (1996)]. The TEM measurements by Sugawara showed that when the aluminum film was deposited between 80 nm and 200 nm thick, grains with micrometers in size were formed. Because the coalescence of the island structures improved the orientation of the grains, TEM showed single crystal-like diffraction peaks, consistent with what we observed with the MeV electrons. Therefore, our single-crystal samples were made of highly oriented micrometer-sized grains.

From reviewer's report:

2. The shock loading is carried out via ablation using a 20 ps and 10 mJ laser. The interaction of Al with a laser that involves ablation is likely to result in the melting of the metal before the generation of the shock wave. In addition, temperatures generated are significantly higher in the shock front. There is no discussion of laser melting and the role of temperatures on the transverse and longitudinal strains.

Response: We agree with the reviewer that the laser ablation results in the melting (and removal) of the metal material before the shock wave generation. However, this process is mostly constrained in the skin depth of the optical light. For 800 nm light, the skin depth in aluminum is about 20 nm. In our MD simulations, we approximated the melting process by excluding the first 20nm layer from the calculation of elastic strains. Within our probed time window, this approximation is reasonable since our experimental results did not show significant melting of the sample (See our response to Comment 6 from the reviewer). Furthermore, our experimental observation is also consistent with what was found from a recent LCLS experiment (1 μ m thick copper ablated by 800 nm ps lasers with intensities of $\sim 10^{11}$ W/cm²) in which the small angle x-ray scattering (SAXS) did not show significant loss of sample during the full compression [J. Coakley et al. Science Advances 6, eabb4434 (2020)]. Note, Coakley et al. provided an estimate of ~ 10 nm for the loss of the sample thickness due to the melting of the ablation process. This supports our approximation of 20nm thickness for the ablation process.

Regarding the comment on the temperature, we can estimate its effect from the perspective of thermal diffusion. Following the flash method of determining thermal diffusivity by Parker et al [W.J. Parker et al. JAP 32, 1679 (1960)], the time (Δt) for the conduction of heat through the full sample thickness (L) is given by:

$$\Delta t = \frac{\omega L^2}{\pi^2 \alpha}$$

where ω is a dimensionless parameter and α is thermal diffusivity in unit of cm^2/s . Substituting $\omega = 6$ which corresponds to the rear surface reaching the maximum temperature, $\alpha = 0.9 \text{ cm}^2/\text{s}$ for Al and $L = 200 \text{ nm}$ into the above equation, one can obtain $\Delta t \sim 270 \text{ ps}$, which is much longer than our probed time window of 70 ps . Therefore, we expect the temperature rise of the overall sample to be small during our probed time window. Nonetheless, in response to Comment 4 from the reviewer, we have performed additional MD simulations with different initial temperatures of 300 K , 600 K and 900 K . The results show the temperature effect will not alter the conclusion of this work.

To resolve concerns in this comment, we added a section named “Effect of laser-induced melting and heating on dynamic compression” (Page 12) in the Supplementary Information.

From reviewer's report:

3. The strains are evaluated based on the distances between Laue peaks. An elastic-plastic transition is suggested at a fluence of 4.3 J/cm^2 and the co-existence of the longitudinal and transverse strains is identified as an indicator of plasticity and a two-wave structure of the shock wave. However, the understanding of a two-wave structure is very clear in the literature.

Response: We agree with the reviewer that two-wave structure has been studied in the literature, which we acknowledged in the original manuscript. However, we believe that the novelty and significance of this work is the first experimental measurement of the incipient plasticity using the ultrafast MeV-UED instrument. Measuring these ultrafast structural dynamical processes at high strain rate conditions has been challenging due to the experimental limitations on spatial and temporal resolution. Our study on the lattice response of single-crystal aluminum provides a complete time history of the incipient plasticity, enabling the determination of the dislocation nucleation and transport processes. So far, such quantitative measurements are not yet available in the literature.

From reviewer's report:

4. The inference of dislocation densities based on strains is observed to peak during stage II. The prediction of dislocation densities based on strains does not account for the temperature-dependent aspects of plasticity and not based on the broadening of the peaks.

Response: To comply with the reviewer, we have performed additional MD simulations to study the effect of the temperature on the plasticity. Since we don't expect significant heating of the overall sample during the compression (See our responses to Comments 2 and 6), we selected two temperatures of 600 K and 900 K as the initial bulk sample temperatures and applied the preheated sample with the same loading profile as the room temperature (RT) case. The evolution of elastic strain for these two temperatures are compared with that of RT condition, which is shown in Figure R8. The results show that with the temperature increases, the transverse strain increases while the

longitudinal strain decreases. However, the overall trends of longitudinal and transverse strains including the onset of plasticity remain similar to those of the RT condition. On the other hand, the results for RT condition show the closest agreement with UED data, implying a minor effect from the thermal heating of the sample in the time scale of experiment.

Regarding the broadening of the peaks, in addition to dislocation defects, there are many other factors that could contribute to this, for instance strain gradient and residual stress. Compared with polycrystalline samples that have homogenous crystallite distribution, it would be much difficult to determine the dislocation defects from the peak broadening in single crystals. For this reason, we applied the widely used Orowan's equation to estimate the mobile dislocation density.

Figure R8: MD simulated strain evolution in laser-driven compressed (110) single-crystal Al at different initial bulk temperatures, i.e., 300 K, 600 K and 900 K. The loading condition is the same for the three temperatures and is derived from hydrodynamics simulations at a pump fluence of 4.3 J cm^{-2} . N denotes normal or longitudinal strain and T denotes transverse strain. The UED data, denoted by N (T)_expt are shown here for comparison.

To resolve the concerns in this comment, we added a section named “Effect of laser-induced melting and heating on dynamic compression” (Page 12) in the Supplementary Information. In this section, the temperature effect on the strain evolution is discussed.

From reviewer's report:

5. The experiments are complemented using MD simulations that use a 1D hydro-code to predict the ramp duration of the piston to mimic shock loading. Again, such a framework does not account for the ablation/melting behavior of the metal before a shock wave is generated. Such effects are related to the absorption of laser energy by electrons and then transfer to the lattice through electron-phonon coupling. Such an MD framework does not mimic the laser loading conditions and the onset of plasticity at 27 ps is an artificial correlation with experiments. The experiments only discuss mobile densities and no correlation with the distribution of The suggestion of the creation of dislocation embryos is not clear.

Response: The reviewer raised the concern of the ablation/melting behavior in affecting our prediction using the two-step simulation approach. Similar concern was expressed by the reviewer in Comments 2 and 6. As we stated in the response to Comment 2, the ablation and melting behavior is mostly constrained in the front 20-nm layer. This is supported by a recent LCLS experiment by Coakley et al [J. Coakley et al. Science Advances 6, eabb4434 (2020)]. In our response to Comment 6, we have also provided the experimental data which does not show significant melting during the dynamic compression. The electron-phonon coupling time in Al is approximately 1 ps [Waldecker, et al. PRX 6, 021003 (2016)], which is much shorter than the laser pulse length. Consequently, electrons and lattice are in time-averaged thermal equilibrium and the heat diffusion into deeper parts of the material takes place on a time scale set by the lattice thermal diffusivity.

Regarding the comment: “*Such an MD framework does not mimic the laser loading conditions and the onset of plasticity at 27 ps is an artificial correlation with experiments.*”, we respectfully disagree with the reviewer. Our simulation approach that combines MD simulations with hydrodynamics simulations has been demonstrated in modelling the dynamic compression driven by picosecond laser ablation, see the studies led by Milathianaki et al. [D. Milanthianaki, et al. Science 342, 220 (2013) and J. Coakley et al. Science Advances 6, eabb4434 (2020)]. The onset of plasticity at 27 ps is not an artificial correlation with experiments; instead, this is because the elastic limit is reached at this time delay.

Regarding the last comment “*The experiments only discuss mobile densities and no correlation with the distribution of The suggestion of the creation of dislocation embryos is not clear.*”, we would like to stress that transmission-geometry diffraction patterns taken in the stroboscopic manner encodes the structural information from the ensemble average of the compressed sample, as such it does not provide information on the spatial distribution of dislocations. On the other hand, molecular dynamics simulations play an important role in revealing the spatial distribution of dislocation evolution, as shown by Figure 4C of the main text. Regarding dislocation embryos, this is indicated and resolved by the evolution of the dislocation network shown in Figure 4B. At 27 ps, the atoms with large shear strain are shown and dislocations will nucleate randomly at these positions, implying the dislocation incubation.

To resolve the concerns in this comment, we made the following changes in the revised manuscript and the Supplementary Information.

1. Added the following sentence in “Molecular Dynamics Simulation” (Page 14) of the manuscript: “These simulations were based on a two-step simulation method, shown in Supplementary Figure 10, which has been demonstrated in modelling the dynamic compression driven by picosecond laser ablation³⁴.” (Ref 34: J. Coakley et al. Science Advances 6, eabb4434 (2020))
2. Added a section named “Effect of laser-induced melting and heating on dynamic compression” (Page 12) in the supplementary information to discuss the influence from the laser-induced melting.

From reviewer's report:

6. The sample dimensions of 200 nm are very small to get enough insights into the elastic-plastic transitions during laser-shock loading. The phenomena of ablation and melting are likely to consume a good fraction of the metal during the generation of the shock wave. Why are the effects of the solid-liquid interface in the microstructure not observed in the diffractograms?

Response: We respectfully disagree with the reviewer. Our experimental data have clearly shown that sample dimensions of 200 nm are sufficiently thick to study the elastic-plastic transitions induced by laser-shock loading. The transit time for the longitudinal sound wave in 200-nm-thick Al is approximately 32 ps. After the shock breakout, there is another ~ 15 ps time window before the compression state is fully released. This implies a total time window of ~ 50 ps for the compression state to hold, as clearly shown by the evolution of the longitudinal elastic strain in Figure 3 of the main text. Under our experimental conditions, the elastic-plastic transition is observed to take place at around 27 ps. This retarded time provides not only sufficient time for separating the elastic and plastic waves, but also leaves a detection window of ~ 23 ps, which is much greater than the electron pulse length of 0.6 ps, for studying the dislocation dynamics before the full release of the compression state.

It is also worth noting that similar sample dimensions of Al have been previously employed to study elastic-plastic transitions by laser-shock loading [B. Zuanetti et al. JAP 123, 195104 (2018)]. In this experiment, the elastic-plastic response of the Al was measured by a velocimetry technique called ultrafast dynamic ellipsometry and the thinnest sample under studied was 278 nm. While the elastic precursor was not discernible for the 278 nm sample due to the instrumental resolution (~ 20 ps) of the technique, this work demonstrated that sample dimensions of 100s' nm can be served for elastic-plastic transition studies.

The effect from ablation and melting have been addressed in our response to Comment 2 from the reviewer. Our experimental data did not show significant melting of the sample during the probed time window since there is no obvious liquid scattering signal from the diffractograms. Note that liquid scattering signal is featured by a single broad ring in the diffraction pattern with its peak position determined by the liquid mass density [B. Siwick, et al. Science 302, 1382 (2003) and Chem. Phys. 299, 285 (2004), and Mo et al. Science 360, 1451 (2018)]. To support this, in Figure R9 (A), we are providing an example of the radially average intensity lineout of the diffraction data taken at 42 ps after the laser arrival for the highest fluence of 4.3 J cm^{-2} , together with its static reference lineout. The raw diffraction pattern of the 42 ps data is shown in Figure 2 of the main text.

Figure R9. (A) Radially average intensity lineout of the diffraction data at 42 ps (blue line) for the highest fluence of 4.3 J cm^{-2} . Static reference of the diffraction data taken without laser irradiation

was shown by the grey line with the observed diffraction peaks labelled by corresponding Miller indices. What also shown are the predictions of the MeV electron scattering signal for liquid Al (purple line) at temperature of 943 K and the total scattering signal (red line) in which the fitted background signal of the 42 ps diffraction data was added to the liquid scattering signal. The liquid scattering signal was calculated by using the liquid structure factor (black dash-circle line) and electron atomic form factor (red line) as shown in (B). See text for more details.

Comparing with the static reference, the 42 ps data shows that the diffraction peak (220) is remained strong despite of its broadening due to dynamic compression. This implies that there remains a significant fraction of solid with the compressed sample at this delay. Note that the increase of the global background is attributed to the thermal diffuse scattering and the plasma-induced background [B. Siwick, et al. Chem. Phys. 299, 285 (2004) and Mo. et al. Science 360, 1451 (2018)].

Furthermore, in Figure R9 (A), we are also providing the calculation result of the liquid scattering signal (purple line) for liquid aluminum at an equilibrium temperature of 943 K. As indicated, there is a broad liquid ring that peaks at $\sim 2.6 \text{ \AA}^{-1}$. Electron scattering intensity, $I(Q)$, is calculated using the following expression [B. Siwick, et al. Chem. Phys. 299, 285 (2004)]: $I(Q) = F(Q) \cdot f(Q)^2$, where $F(Q)$ and $f(Q)$ are the structure factor and atomic form factor of the material, respectively. In our calculation, we adopted the liquid structure factor (LSF) data of aluminum from Waseda [Y. Waseda, The Structure of Non-Crystalline Materials (McGraw-Hill, New York, 1980).] LSF at temperature of 943 K (Fig. R9 (B)) was chosen because we don't expect significant heating nor superheating for our sample during the dynamic compression. $f(Q)$, shown by the red curve in Fig. R9 (B), was taken from the textbook of Peng [L.M. Peng, et al. High Energy Electron Diffraction and Microscopy, 2004].

To directly compare with the measured scattering signal, we added to the calculated liquid scattering data with the fitted global background [B. Siwick, et al. Chem. Phys. 299, 285 (2004)] from the 42 ps data, the result of which is shown by the red curve in Fig. R9 (A). The comparison is clear that the 42 ps data does not have the liquid peak as expected for liquid aluminum, which further corroborates that the sample remains mostly solid at this time delay.

From reviewer's report:

7. The generation of plasticity through dislocations is likely to create a high density of stacking faults in the microstructure. Would these stacking faults with an hcp structure provide additional peaks in the diffractograms?

Response: To comply with the reviewer, we have performed X-ray diffraction simulations of the compressed sample to see if there will be additional peaks in the diffractograms due to the stacking faults. Representative results for the highest fluence of 4.3 J cm^{-2} are shown in Figure R10 with the two delays of 0 ps and 42 ps. Note that the volume fraction of stacking faults in the 42 ps data is about 8.9%. However, there are no additional peaks in the simulation pattern, which means there will not be additional peaks in our electron diffractograms.

Figure R10: XRD simulations of Al-[110] for pump fluence of 4.3 J cm^{-2} at 0 ps and 42 ps.

To resolve the concern in this comment, we added a section named “Effect of stacking faults on diffraction peaks” (Page 10) in the Supplementary Information of our revised manuscript.

From reviewer's report:

There are many more challenges in the interpretation of the results discussed in this manuscript given the small dimensions of the Al sample, generation of the shock using a laser, and the interpretation of dislocation densities based only on strains using diffractograms. The experimental data and the modeling data are two disconnected discussions in the manuscript.

Nonetheless, there are no new insights that improve our understanding of the deformation response of fcc metals. The manuscript is therefore not recommended for publication.

Response: The reviewer pointed out the challenges of this study and the drawbacks of our original manuscript. For every single one of them, we have provided a comprehensive response and modified our manuscript accordingly. We believe that the quality of our revised manuscript has been improved significantly.

As we stated earlier in the response, the novelty and significance of this work is the first experimental measurement of the incipient plasticity at high strain rate conditions and the accurate determination of the dislocation dynamics. Such quantitative measurements are not yet available in the literature. We expect our work to be of interest to a broad audience in dynamic compression.

REVIEWER COMMENTS

Reviewer #1 (Remarks to the Author):

This paper presents direct real-time spatially resolved data about the initiation of plasticity. Both the spatial resolution and the time resolution are adequate to compare to theories of plasticity. The work is technically challenging but provides first of their kind data. The authors have adequately addressed all the issues I had with the paper. The work is well described. It is an important contribution to the plasticity literature and deserves to be published.

Reviewer #2 (Remarks to the Author):

The authors adequately addressed the critical comments from me and other reviewers and produced an enhanced version which I now recommend to accept for publication.

Reviewer #3 (Remarks to the Author):

The authors have suggested that the novelty is the first experimental measurement of incipient plasticity. While the authors do demonstrate this capability, the concern is the lack of new insights from these experiments.

The experimental contributions here are the temporal evolution of diffractograms and strains in time domains of several picoseconds that can be very valuable for validation of MD simulations.

The hydrodynamic simulations are used to investigate the laser interactions and MD simulations to extract plasticity contributors.

While the authors justify use of piston shock to reproduce stress states in MD simulations, the laser-matter interactions modifies the shock structure generated in MD simulations. The added shock structure profiles (Figure R7) is far from what is observed under laser shock conditions. The authors should consider shock structures generated under laser shock conditions [<https://aip.scitation.org/doi/10.1063/1.5051618>].

The study also does not demonstrate any peak shifts due to presence of stacking faults. Peak shifts have been reported in the literature [<https://doi.org/10.1107/S0021889800000133>]; [<https://journals.aps.org/prx/pdf/10.1103/PhysRevX.10.011010>]

Thus, while the capability is new and exciting, the manuscript fails to provide any new quantitative insights in the shock response of the microstructure. The MD simulations oversimplify these processes of shock generation.

The manuscript is still not recommended for publication.

REVIEWER COMMENTS**Response to Reviewer 1:**

From reviewer's report:

This paper presents direct real-time spatially resolved data about the initiation of plasticity. Both the spatial resolution and the time resolution are adequate to compare to theories of plasticity. The work is technically challenging but provides first of their kind data. The authors have adequately addressed all the issues I had with the paper. The work is well described. It is an important contribution to the plasticity literature and deserves to be published.

Response:

We thank the reviewer for recognizing the importance and the technical challenge of this work. We are pleased that the reviewer is satisfied with our revision. We thank the reviewer again for helping with improving our manuscript.

Response to Reviewer 2:

From reviewer's report:

The authors adequately addressed the critical comments from me and other reviewers and produced an enhanced version which I now recommend to accept for publication.

Response:

We are happy that all critical comments and concerns from the reviewer and other reviewers are adequately resolved. We thank the reviewer again for helping with improving our manuscript.

Response to Reviewer 3:

We thank the reviewer for additional comments that helped us improve the quality of our manuscript. We have addressed all the concerns from the reviewer and made changes to our manuscript accordingly. Our detailed responses are found as follows.

Reviewer #3 (Remarks to the Author):

From reviewer's report:

The authors have suggested that the novelty is the first experimental measurement of incipient plasticity. While the authors do demonstrate this capability, the concern is the lack of new insights from these experiments.

The experimental contributions here are the temporal evolution of diffractograms and strains in time domains of several picoseconds that can be very valuable for validation of MD simulations.

The hydrodynamic simulations are used to investigate the laser interactions and MD simulations to extract plasticity contributors.

While the authors justify use of piston shock to reproduce stress states in MD simulations, the laser-matter interactions modifies the shock structure generated in MD simulations. The added shock structure profiles (Figure R7) is far from what is observed under laser shock conditions. The authors should consider shock structures generated under laser shock conditions [<https://aip.scitation.org/doi/10.1063/1.5051618>].

Response:

We agree with the reviewer that laser-matter interactions can affect the shock structure. However, an accurate incorporation of laser-matter interaction physics in laser-driven dynamic compression simulation is technically challenging for the whole community. In the paper suggested by the reviewer [Galitskiy et al. J. Appl. Phys. 124, 205901(2018)], two-temperature model molecular dynamics (TTM-MD) simulation was employed to study femtosecond laser-induced shock compression. However, the validity of TTM model in approximating femtosecond laser energy deposition has been challenged by a nonthermal lattice model [L. Waldecker et al. PRX 6, 021003 (2016)]. In picosecond laser-matter interaction, laser energy deposition is complicated by the concurrent plasma expansion process and is better described by hydrodynamic codes [R. Ramis et al. Comput. Phys. Commun. 183, 637 (2012)]. Given that a comprehensive full-scale simulation is not practical or even possible, we followed previous studies [D. Milathianaki, et al. Science 342, 220 (2013) and J. Coakley et al. Science Advances 6, eabb4434 (2020)], and adopted the two-step simulation approach to model the dynamic compression induced by picosecond laser ablation.

To address this concern from reviewer, we made the following changes in the “MD simulations” section (Page 14) of Method in the manuscript to further clarify our two-step simulation approach.

Changed from: “We performed molecular dynamics (MD) simulations of the transverse and longitudinal elastic strain evolutions in single-crystal Al undergoing laser-driven dynamic compression. These simulations were based on a two-step simulation method, shown in Supplementary Figure 10, which has been demonstrated in modelling the dynamic compression driven by picosecond laser ablation³⁴.” [Ref. 34 : J. Coakley et al. Science Advances 6, eabb4434 (2020)]

to: “We performed molecular dynamics (MD) simulations of the transverse and longitudinal elastic strain evolutions in single-crystal Al undergoing laser-driven dynamic compression. **Given that a comprehensive full-scale simulation is challenging, we adopted a two-step simulation, shown in Supplementary Figure 10, to provide a physical understanding of the key experimental results. This simulation method has been demonstrated in modelling the dynamic compression driven by picosecond laser ablation³⁴.**”

From reviewer's report:

The study also does not demonstrate any peak shifts due to presence of stacking faults. Peak shifts have been reported in the literature [<https://doi.org/10.1107/S0021889800000133>]; [<https://journals.aps.org/prx/pdf/10.1103/PhysRevX.10.011010>]

Response:

A detailed discussion on the stacking fault effect on the peak shift was provided in our previous response to Comment 2 from Reviewer #2. In addition, we also made corresponding changes in our revised manuscript (Page 8) to address the same concern, supplemented by a specified section (Page 10) in our revised supplementary information. Here, we will provide a brief discussion why the stacking faults won't affect the peak shifts in our measurement.

Following the same analysis as described in Warren's textbook and in the work by Sharma et.al. [Sharma et al., PRX 10, 11010 (2020)] (the same reference as pointed out by the reviewer), the {220}-family peaks measured in our experiment follow the law $|L_0| = |h + k + l| = 0$, indicating that they belong to the un-affected components. Furthermore, we performed additional MD simulations to verify this, and the results show no peak shift due to stacking faults under our experimental conditions. The simulations results are shown in Figure R3 in the previous response letter, also in Supplementary Figure 11 (B) of the revised supplementary information.

From reviewer's report:

Thus, while the capability is new and exciting, the manuscript fails to provide any new quantitative insights in the shock response of the microstructure. The MD simulations oversimplify these processes of shock generation.

The manuscript is still not recommended for publication.

Response:

We thank the reviewer for acknowledging the novelty of this experiment. However, we respectfully disagree with the reviewer that our work provides no new insights. Our experiment provides first of its kind data to visualize the incipient plasticity at high strain rate conditions, which allows the direct determination of dislocation nucleation and transport processes in plastic deformation. These are new insights and are independent of MD simulations. The goal of our MD simulations is to help provide a physical understanding of the key experimental results. Lastly, this demonstrated time-resolved diffraction technique using MeV electrons would open a new horizon for investigating a broad range of high-pressure and high strain-rate phenomena. We expect our work to be of interest to a broad audience in dynamic compression.

REVIEWERS' COMMENTS

Reviewer #3 (Remarks to the Author):

The authors have addressed all the comments in detail. The manuscript is recommended for publication.